# CONDITIONAL CONTRASTIVE LEARNING WITH KERNEL

**Yao-Hung Hubert Tsai**[1†], **Tianqin Li**[1†], **Martin Q. Ma**[1], **Han Zhao**[2],
**Kun Zhang**[1,3], **Louis-Philippe Morency**[1], **& Ruslan Salakhutdinov**[1]
[1]Carnegie Mellon University [2]University of Illinois at Urbana-Champaign
[3]Mohamed bin Zayed University of Artificial Intelligence
`{yaohungt, tianqinl, qianlim, kunz1, morency, rsalakhu}@cs.cmu.edu`
`{hanzhao}@illinois.edu`

## ABSTRACT

Conditional contrastive learning frameworks consider the conditional sampling procedure that constructs positive or negative data pairs conditioned on specific variables. Fair contrastive learning constructs negative pairs, for example, from the same gender (conditioning on sensitive information), which in turn reduces undesirable information from the learned representations; weakly supervised contrastive learning constructs positive pairs with similar annotative attributes (conditioning on auxiliary information), which in turn are incorporated into the representations. Although conditional contrastive learning enables many applications, the conditional sampling procedure can be challenging if we cannot obtain sufficient data pairs for some values of the conditioning variable. This paper presents Conditional Contrastive Learning with Kernel (CCL-K) that converts existing conditional contrastive objectives into alternative forms that mitigate the insufficient data problem. Instead of sampling data according to the value of the conditioning variable, CCL-K uses the Kernel Conditional Embedding Operator that samples data from all available data and assigns weights to each sampled data given the kernel similarity between the values of the conditioning variable. We conduct experiments using weakly supervised, fair, and hard negatives contrastive learning, showing CCL-K outperforms state-of-the-art baselines.

## 1 INTRODUCTION

Contrastive learning algorithms (Oord et al., 2018; Chen et al., 2020; He et al., 2020; Khosla et al., 2020) learn similar representations for *positively-paired* data and dissimilar representations for *negatively-paired* data. For instance, self-supervised visual contrastive learning (Hjelm et al., 2018) define two views of the same image (applying different image augmentations to each view) as a positive pair and different images as a negative pair. Supervised contrastive learning (Khosla et al., 2020) defines data with the same labels as a positive pair and data with different labels as a negative pair. We see that distinct contrastive approaches consider different positive pairs and negative pairs constructions according to their learning goals.

In conditional contrastive learning, positive and negative pairs are constructed conditioned on specific variables. The conditioning variables can be downstream labels (Khosla et al., 2020), sensitive attributes (Tsai et al., 2021c), auxiliary information (Tsai et al., 2021a), or data embedding features (Robinson et al., 2020; Wu et al., 2020). For example, in fair contrastive learning (Tsai et al., 2021c), conditioning on variables such as gender or race, is performed to remove undesirable information from the learned representations. Conditioning is achieved by constructing negative pairs from the same gender. As a second example, in weakly-supervised contrastive learning (Tsai et al., 2021a), the aim is to include extra information in the learned representations. This extra information could be, for example, some freely available attributes for images collected from social media. The conditioning is performed by constructing positive pairs with similar annotative attributes. The cornerstone of conditional contrastive learning is the *conditional sampling procedure*: efficiently constructing positive or negative pairs while properly enforcing conditioning.

---

[†]Equal contribution. Code available at: `https://github.com/Crazy-Jack/CCLK-release`.

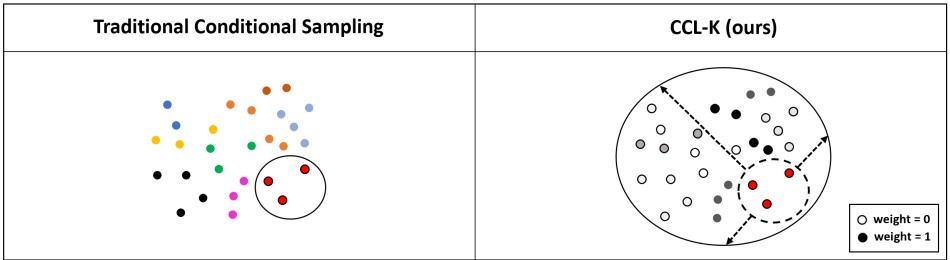

Figure 1: Illustration of the main idea in CCL-K, best viewed in color. Suppose we select color as the conditioning variable and we want to sample red data points. **Left figure:** The traditional conditional sampling procedure only samples red points (i.e., the points in the circle). **Right figure:** The proposed CCL-K samples all data points (i.e., the sampled set expands from the inner circle to the outer circle) with a weighting scheme based on the similarities between conditioning variables' values. The higher the weight, the higher probability of of a data point being sampled. For example, CCL-K can sample orange data with a high probability, because orange resembles to red. In this illustration, the weight ranges from 0 to 1 with white as 0 and black as 1.

The conditional sampling procedure requires access to sufficient data for each state of the conditioning variables. For example, if we are conditioning on the "age" attribute to reduce the age bias, then the conditional sampling procedure will work best if we can create a sufficient number of data pairs for each age group. However, in many real-world situations, some values of the conditioning variable may not have enough data points or even no data points at all. The sampling problem exacerbates when the conditioning variable is continuous.

In this paper, we introduce Conditional Contrastive Learning with Kernel (CCL-K), to help mitigate the problem of insufficient data, by providing an alternative formulation using similarity kernels (see Figure 1). Given a specific value of the conditioning variable, instead of sampling data that are exactly associated with this specific value, we can also sample data that have similar values of the conditioning variable. We leverage the Kernel Conditional Embedding Operator (Song et al., 2013) for the sampling process, which considers kernels (Schölkopf et al., 2002) to measure the similarities between the values of the conditioning variable. This new formulation with a weighting scheme based on similarity kernel allows us to use all training data when conditionally creating positive and negative pairs. It also enables contrastive learning to use continuous conditioning variables.

To study the generalization of CCL-K, we conduct experiments on three tasks. The first task is weakly supervised contrastive learning , which incorporates auxiliary information by conditioning on the annotative attribute to improve the downstream task performance. For the second task, fair contrastive learning , we condition on the sensitive attribute to remove its information in the representations. The last task is hard negative contrastive learning , which samples negative pairs to learn dissimilar representations where the negative pairs have similar outcomes from the conditioning variable. We compare CCL-K with the baselines tailored for each of the three tasks, and CCL-K outperforms all baselines on downstream evaluations.

## 2 CONDITIONAL SAMPLING IN CONTRASTIVE LEARNING

In Section 2.1, we first present the technical preliminaries of contrastive learning. Next, we introduce the *conditional sampling procedure* in Section 2.2, showing that it instantiates recent conditional contrastive frameworks. Last, in Section 2.3, we discuss the limitation of insufficient data in the current framework, presenting to convert existing objectives into alternative forms with kernels to alleviate the limitation. In the paper, we use uppercase letters (e.g., $X$) for random variables, $P._{\cdot}$ for the distribution (e.g., $P_X$ denotes the distribution of $X$), lowercase letters (e.g., $x$) for the outcome from a variable, and the calligraphy letter (e.g., $\mathcal{X}$) for the sample space of a variable.

### 2.1 TECHNICAL PRELIMINARIES - UNCONDITIONAL CONTRASTIVE LEARNING

Contrastive methods learn similar representations for positive pairs and dissimilar representations for negative pairs (Chen et al., 2020; He et al., 2020; Hjelm et al., 2018). In prior literature (Oord et al., 2018; Tsai et al., 2021b; Bachman et al., 2019; Hjelm et al., 2018), the construction of the positive and negative pairs can be understood as sampling from the joint distribution $P_{XY}$ and product of marginal $P_X P_Y$. To see this, we begin by reviewing one popular contrastive approach, the InfoNCE objective (Oord et al., 2018):

$$\text{InfoNCE} := \sup_f \mathbb{E}_{(x,y_{\text{pos}}) \sim P_{XY}, \{y_{\text{neg,i}}\}_{i=1}^n \sim P_Y^{\otimes n}} \left[ \log \frac{e^{f(x,y_{\text{pos}})}}{e^{f(x,y_{\text{pos}})} + \sum_{i=1}^n e^{f(x,y_{\text{neg,i}})}} \right], \quad (1)$$

| Framework | Conditioning Variable $Z$ | Positive Pairs from | Negative Pairs from |
|---|---|---|---|
| Weakly Supervised (Tsai et al., 2021a) | Auxiliary information | $P_{X\|Z=z}P_{Y\|Z=z}$ | $P_X P_Y$ |
| Fair (Tsai et al., 2021c) | Sensitive information | $P_{XY\|Z=z}$ | $P_{X\|Z=z}P_{Y\|Z=z}$ |
| Hard Negatives (Wu et al., 2020) | Feature embedding of $X$ | $P_{XY}$ | $P_{X\|Z=z}P_{Y\|Z=z}$ |

Table 1: Conditional sampling procedure of weakly supervised, fair, and hard negative contrastive learning. We can regard the data pair $(x, y)$ sampled from $P_{XY}$ or $P_{XY|Z}$ as strongly-correlated, such as views of the same image by applying different image augmentations; $(x, y)$ sampled from $P_{X|Z}P_{Y|Z}$ as two random data that are both associated with the same outcome of the conditioning variable, such as two random images with the same annotative attributes; and $(x, y)$ from $P_X P_Y$ as two uncorrelated data such as two random images.

where $X$ and $Y$ represent the data and $x$ is the anchor data. $(x, y_{\mathrm{pos}})$ are positively-paired and are sampled from $P_{XY}$ ($x$ and $y$ are different views to each other; e.g., augmented variants of the same image), and $\{(x, y_{\mathrm{neg,i}})\}_{i=1}^n$ are negatively-paired and are sampled from $P_X P_Y$ (e.g., $x$ and $y$ are two random images). $f(\cdot, \cdot)$ defines a mapping $\mathcal{X} \times \mathcal{Y} \to \mathbb{R}$, which is parameterized via neural nets (Chen et al., 2020) as:

$$f(x, y) := \text{cosine similarity}\Big(g_{\theta_X}(x), g_{\theta_Y}(y)\Big)/\tau, \tag{2}$$

where $g_{\theta_X}(x)$ and $g_{\theta_Y}(y)$ are embedded features, $g_{\theta_X}$ and $g_{\theta_Y}$ are neural networks ($g_{\theta_X}$ can be the same as $g_{\theta_Y}$) parameterized by parameters $\theta_X$ and $\theta_Y$, and $\tau$ is a hyper-parameter that rescales the score from the cosine similarity. The InfoNCE objective aims to maximize the similarity score between a data pair sampled from the joint distribution (i.e., $(x, y_{\mathrm{pos}}) \sim P_{XY}$) and minimize the similarity score between a data pair sampled from the product of marginal distribution (i.e., $(x, y_{\mathrm{neg}}) \sim P_X P_Y$) (Tschannen et al., 2019).

## 2.2 CONDITIONAL CONTRASTIVE LEARNING

Recent literature (Robinson et al., 2020; Tsai et al., 2021a;c) has modified the InfoNCE objective to achieve different learning goals by sampling positive or negative pairs under conditioning variable $Z$ (and its outcome $z$). These different conditional contrastive learning frameworks have one common technical challenge: the conditional sampling procedure. The conditional sampling procedure samples the positive or negative data pairs from the product of conditional distribution: $(x, y) \sim P_{X|Z=z}P_{Y|Z=z}$, where $x$ and $y$ are sampled given the same outcome from the conditioning variable (e.g., two random images with blue sky background, when selecting $z$ as blue sky background). We summarize how different frameworks use the conditional sampling procedure in Table 1.

**Weakly Supervised Contrastive Learning.** Tsai et al. (2021a) consider the auxiliary information from data (e.g., annotation attributes of images) as a weak supervision signal and propose a contrastive objective to incorporate the weak supervision in the representations. This work is motivated by the argument that the auxiliary information implies semantic similarities. With this motivation, the weakly supervised contrastive learning framework learns similar representations for data with the same auxiliary information and dissimilar representations for data with different auxiliary information. Embracing this idea, the original InfoNCE objective can be modified into the weakly supervised InfoNCE (abbreviated as WeaklySup-InfoNCE) objective:

$$\text{WeaklySup}_{\text{InfoNCE}} := \sup_f \mathbb{E}_{z \sim P_Z, (x, y_{\mathrm{pos}}) \sim P_{X|Z=z}P_{Y|Z=z}, \{y_{\mathrm{neg,i}}\}_{i=1}^n \sim P_Y^{\otimes n}} \left[ \log \frac{e^{f(x, y_{\mathrm{pos}})}}{e^{f(x, y_{\mathrm{pos}})} + \sum_{i=1}^n e^{f(x, y_{\mathrm{neg,i}})}} \right]. \tag{3}$$

Here $Z$ is the conditioning variable representing the auxiliary information from data, and $z$ is the outcome of auxiliary information that we sample from $P_Z$. $(x, y_{\mathrm{pos}})$ are positive pairs sampled from $P_{X|Z=z}P_{Y|Z=z}$. In this design, the positive pairs always have the same outcome from the conditioning variable. $\{(x, y_{\mathrm{neg,i}})\}_{i=1}^n$ are negative pairs that are sampled from $P_X P_Y$.

**Fair Contrastive Learning.** Another recent work (Tsai et al., 2021c) presented to remove undesirable sensitive information (such as gender) in the representation, by sampling negative pairs conditioning on sensitive attributes. The paper argues that fixing the outcome of the sensitive variable prevents the model from using the sensitive information to distinguish positive pairs from negative pairs (since all positive and negative samples share the same outcome), and the model will ignore the effect of the sensitive attribute during contrastive learning. Embracing this idea, the original InfoNCE objective can be modified into the Fair-InfoNCE objective:

$$\text{Fair}_{\text{InfoNCE}} := \sup_f \mathbb{E}_{z \sim P_Z, (x, y_{\mathrm{pos}}) \sim P_{XY|Z=z}, \{y_{\mathrm{neg,i}}\}_{i=1}^n \sim P_{Y|Z=z}^{\otimes n}} \left[ \log \frac{e^{f(x, y_{\mathrm{pos}})}}{e^{f(x, y_{\mathrm{pos}})} + \sum_{i=1}^n e^{f(x, y_{\mathrm{neg,i}})}} \right]. \tag{4}$$

Here $Z$ is the conditioning variable representing the sensitive information (e.g., gender), $z$ is the outcome of the sensitive information (e.g., female), and the anchor data $x$ is associated with $z$ (e.g., a

data point that has the gender attribute being female). $(x, y_{\text{pos}})$ are positively-paired that are sampled from $P_{XY|Z=z}$, and $x$ and $y_{\text{pos}}$ are constructed to have the same $z$. $\{(x, y_{\text{neg},i})\}_{i=1}^n$ are negatively-paired that are sampled from $P_{X|Z=z}P_{Y|Z=z}$. In this design, the positively-paired samples and the negatively-paired samples are always having the same outcome from the conditioning variable.

**Hard-negative Contrastive Learning.** Robinson et al. (2020) and Kalantidis et al. (2020) argue that contrastive learning can benefit from hard negative samples (i.e., samples $y$ that are difficult to distinguish from an anchor $x$). Rather than considering two arbitrary data as negatively-paired, these methods construct a negative data pair from two random data that are not too far from each other[1]. Embracing this idea, the original InfoNCE objective is modified into the Hard Negative InfoNCE (abbreviated as HardNeg-InfoNCE) objective:

$$\text{HardNeg}_{\text{InfoNCE}} := \sup_f \mathbb{E}_{(x, y_{\text{pos}}) \sim P_{XY}, \, z \sim P_{Z|X=x}, \, \{y_{\text{neg},i}\}_{i=1}^n \sim P_{Y|Z=z}^{\otimes n}} \left[ \log \frac{e^{f(x, y_{\text{pos}})}}{e^{f(x, y_{\text{pos}})} + \sum_{i=1}^n e^{f(x, y_{\text{neg},i})}} \right]. \quad (5)$$

Here $Z$ is the conditioning variable representing the embedding feature of $X$, in particular $z = g_{\theta_X}(x)$ (see definition in equation 2, we refer $g_{\theta_X}(x)$ as the embedding feature of $x$ and $g_{\theta_Y}(y)$ as the embedding feature of $y$). $(x, y_{\text{pos}})$ are positively-paired that are sampled from $P_{XY}$. To construct negative pairs $\{(x, y_{\text{neg},i})\}_{i=1}^n$, we sample $\{(y_{\text{neg},i})\}_{i=1}^n$ from $P_{Y|Z=z=g_{\theta_X}(x)}$. We realize the sampling from $P_{Y|Z=z=g_{\theta_X}(x)}$ as sampling data points from $Y$ whose embedding features are close to $z = g_{\theta_X}(x)$: sampling $y$ such that $g_{\theta_Y}(y)$ is close to $g_{\theta_X}(x)$.

## 2.3 CONDITIONAL CONTRASTIVE LEARNING WITH KERNEL

The conditional sampling procedure common to all these conditional contrastive frameworks has a limitation when we have insufficient data points associated with some outcomes of the conditioning variable. In particular, given an anchor data $x$ and its corresponding conditioning variable's outcome $z$, if $z$ is uncommon, then it will be challenging for us to sample $y$ that is associated with $z$ via $y \sim P_{Y|Z=z}$[2]. The insufficient data problem can be more serious when the cardinality of the conditioning variable $|Z|$ is large, which happens when $Z$ contains many discrete values, or when $Z$ is a continuous variable (cardinality $|Z| = \infty$). In light of this limitation, we present to convert these objectives into alternative forms that can avoid the need to sample data from $P_{Y|Z}$ and can retain the same functions as the original forms. We name this new family of formulations **C**onditional **C**ontrastive **L**earning with **K**ernel (CCL-K).

**High Level Intuition.** The high level idea of our method is that, instead of sampling $y$ from $P_{Y|Z=z}$, we sample $y$ from existing data of $Y$ whose associated conditioning variable's outcome is close to $z$. For example, assuming the conditioning variable $Z$ to be age and $z$ to be 80 years old, instead of sampling the data points at the age of 80 directly, we sample from all data points, assigning highest weights to the data points at the age from 70 to 90, given their proximity to 80. Our intuition is that data with similar outcomes from the conditioning variable should be used in support of the conditional sampling. Mathematically, instead of sampling from $P_{Y|Z=z}$, we sample from a distribution proportional to the weighted sum $\sum_{j=1}^N w(z_j, z) P_{Y|Z=z_j}$, where $w(z_j, z)$ represents how similar in the space of of the conditioning variable $Z$. This similarity is computed for all data points $j = 1 \cdots N$. In this paper, we use the Kernel Conditional Embedding Operator (Song et al., 2013) for such approximation, where we represent the similarity using kernel (Schölkopf et al., 2002).

***Step I - Problem Setup.*** We want to avoid the conditional sampling procedure from existing conditional learning objectives (equation 3, 4, 5), and hence we are not supposed to have access to data pairs from the conditional distribution $P_{X|Z}P_{Y|Z}$. Instead, the only given data will be a batch of triplets $\{(x_i, y_i, z_i)\}_{i=1}^b$, which are independently sampled from the joint distribution $P_{XYZ}^{\otimes b}$ with $b$ being the batch size. In particular, when $(x_i, y_i, z_i) \sim P_{XYZ}$, $(x_i, y_i)$ is a pair of data sampled from the joint distribution $P_{XY}$ (e.g., two augmented views of the same image) and $z_i$ is the associated conditioning variable's outcome (e.g., the annotative attribute of the image). To convert previous objectives into alternative forms that avoid the need of the conditional sampling procedure, we need to perform an estimation of the scoring function $e^{f(x,y)}$ for $(x, y) \sim P_{X|Z}P_{Y|Z}$ in equation 3, 4, 5 given only $\{(x_i, y_i, z_i)\}_{i=1}^b \sim P_{XYZ}^{\otimes b}$.

---

[1]Wu et al. (2020) argues that a better construction of negative data pairs is selecting two random data that are neither too far nor too close to each other.

[2]If $x$ is the anchor data and $z$ is its corresponding variable's outcome, then for $y \sim P_{Y|Z=z}$, the data pair $(x, y)$ can be seen as sampling from $P_{X|Z=z}P_{Y|Z=z}$.

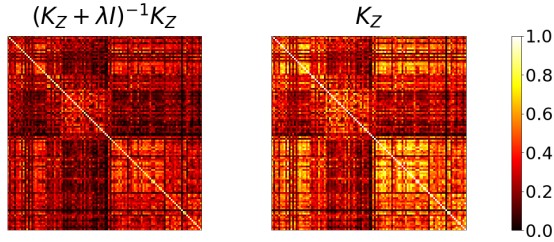

Figure 2: $(K_Z + \lambda I)^{-1}K_Z$ v.s. $K_Z$. We apply min-max normalization $(x - \min(x))/(\max(x) - \min(x))$ for both matrices for better visualization. We see that $(K_Z + \lambda I)^{-1}K_Z$ can be seen as a smoothed version of $K_Z$, which suggests that each entry in $(K_Z + \lambda I)^{-1}K_Z$ represents the similarities between $z$s.

***Step II - Kernel Formulation.*** We present to reformulate previous objectives into kernel expressions. We denote $K_{XY} \in \mathbb{R}^{b \times b}$ as a kernel gram matrix between $X$ and $Y$: let the $i_{\text{th}}$ row and $j_{\text{th}}$ column of $K_{XY}$ be the exponential scoring function $e^{f(x_i, y_j)}$ (see equation 2) with $[K_{XY}]_{ij} := e^{f(x_i, y_j)}$. $K_{XY}$ is a gram matrix because the design of $e^{f(x,y)}$ satisfies a kernel[3] between $g_{\theta_X}(x)$ and $g_{\theta_Y}(y)$:

$$e^{f(x,y)} = \exp\left(\text{cosine similarity}(g_{\theta_X}(x), g_{\theta_Y}(y))/\tau\right) := \left\langle \phi(g_{\theta_X}(x)), \phi(g_{\theta_Y}(y)) \right\rangle_{\mathcal{H}}, \quad (6)$$

where $\langle \cdot, \cdot \rangle_{\mathcal{H}}$ is the inner product in a Reproducing Kernel Hilbert Space (RKHS) $\mathcal{H}$ and $\phi$ is the corresponding feature map. $K_{XY}$ can also be represented as $K_{XY} = \Phi_X \Phi_Y^\top$ with $\Phi_X = \left[\phi(g_{\theta_X}(x_1)), \cdots, \phi(g_{\theta_X}(x_b))\right]^\top$ and $\Phi_Y = \left[\phi(g_{\theta_Y}(y_1)), \cdots, \phi(g_{\theta_Y}(y_b))\right]^\top$. Similarly, we denote $K_Z \in \mathbb{R}^{b \times b}$ as a kernel gram matrix for $Z$, where $[K_Z]_{ij}$ represents the similarity between $z_i$ and $z_j$: $[K_Z]_{ij} := \left\langle \gamma(z_i), \gamma(z_j) \right\rangle_{\mathcal{G}}$, where $\gamma(\cdot)$ is an arbitrary kernel embedding for $Z$ and $\mathcal{G}$ is its corresponding RKHS space. $K_Z$ can also be represented as $K_Z = \Gamma_Z \Gamma_Z^\top$ with $\Gamma_Z = \left[\gamma(z_1), \cdots, \gamma(z_b)\right]^\top$.

***Step III - Kernel-based Scoring Function $e^{f(x,y)}$ Estimation.*** We present the following:

**Definition 2.1** (Kernel Conditional Embedding Operator (Song et al., 2013))**.** *By Kernel Conditional Embedding Operator (Song et al., 2013), the finite-sample kernel estimation of* $\mathbb{E}_{y \sim P_{Y|Z=z}}\left[\phi(g_{\theta_Y}(y))\right]$ *is* $\Phi_Y^\top(K_Z + \lambda \mathbf{I})^{-1}\Gamma_Z \gamma(z)$*, where* $\lambda$ *is a hyper-parameter.*

**Proposition 2.2** (Estimation of $e^{f(x_i, y)}$ when $y \sim P_{Y|Z=z_i}$)**.** *Given* $\{(x_i, y_i, z_i)\}_{i=1}^b \sim P_{XYZ}^{\otimes b}$*, the finite-sample kernel estimation of* $e^{f(x_i, y)}$ *when* $y \sim P_{Y|Z=z_i}$ *is* $\left[K_{XY}(K_Z + \lambda \mathbf{I})^{-1}K_Z\right]_{ii}$*.* $\left[K_{XY}(K_Z + \lambda \mathbf{I})^{-1}K_Z\right]_{ii} = \sum_{j=1}^b w(z_j, z_i) \, e^{f(x_i, y_j)}$ *with* $w(z_j, z_i) = \left[(K_Z + \lambda \mathbf{I})^{-1}K_Z\right]_{ji}$*.*

*Proof.* For any $Z = z$, along with Definition 2.1, we estimate $\phi(g_{\theta_Y}(y))$ when $y \sim P_{Y|Z=z} \approx \mathbb{E}_{y \sim P_{Y|Z=z}}\left[\phi(g_{\theta_Y}(y))\right] \approx \Phi_Y^\top(K_Z + \lambda \mathbf{I})^{-1}\Gamma_Z \, \gamma(z)$. Then, we plug in the result for the data pair $(x_i, z_i)$ to estimate $e^{f(x_i, y)}$ when $y \sim P_{Y|Z=z_i}$:

$\left\langle \phi(g_{\theta_X}(x_i)), \Phi_Y^\top(K_Z + \lambda \mathbf{I})^{-1}\Gamma_Z \, \gamma(z_i) \right\rangle_{\mathcal{H}} = \text{tr}(\phi(g_{\theta_X}(x_i))^\top \Phi_Y^\top(K_Z + \lambda \mathbf{I})^{-1}\Gamma_Z \, \gamma(z_i)) = [K_{XY}]_{i*}(K_Z + \lambda \mathbf{I})^{-1}[K_Z]_{*i} = [K_{XY}]_{i*}\left[(K_Z + \lambda \mathbf{I})^{-1}K_Z\right]_{*i} = \left[K_{XY}(K_Z + \lambda \mathbf{I})^{-1}K_Z\right]_{ii}. \qquad \square$

$[K_{XY}(K_Z + \lambda \mathbf{I})^{-1}K_Z]_{ii}$ is the kernel estimation of $e^{f(x_i, y)}$ when $(x_i, z_i) \sim P_{XZ}$, $y \sim P_{Y|Z=z_i}$. It defines the similarity between the data pair sampled from $P_{X|Z=z_i}P_{Y|Z=z_i}$. Hence, $(K_Z + \lambda \mathbf{I})^{-1}K_Z$ can be seen as a transformation applied on $K_{XY}$ ($K_{XY}$ defines the similarity between $X$ and $Y$), converting *unconditional* to *conditional* similarities between $X$ and $Y$ (conditioning on $Z$). Proposition 2.2 also re-writes this estimation as a weighted sum over $\{e^{f(x_i, y_j)}\}_{j=1}^n$ with the weights $w(z_j, z_i) = \left[(K_Z + \lambda \mathbf{I})^{-1}K_Z\right]_{ji}$. We provide an illustration to compare $(K_Z + \lambda \mathbf{I})^{-1}K_Z$ and $K_Z$ in Figure 2, showing that $(K_Z + \lambda \mathbf{I})^{-1}K_Z$ can be seen as a smoothed version of $K_Z$, suggesting the weight $\left[(K_Z + \lambda \mathbf{I})^{-1}K_Z\right]_{ji}$ captures the similarity between $(z_j, z_i)$. To conclude, we use the Kernel Conditional Embedding Operator (Song et al., 2013) to avoid explicitly sampling $y \sim P_{Y|Z=z}$, which alleviates the limitation of having insufficient data from $Y$ that are associated with $z$. It is worth noting that our method neither generates raw data directly nor includes additional training.

---

[3]Cosine similarity is a proper kernel, and the exponential of a proper kernel is also a proper kernel.

In terms of computational complexity, calculating the inverse $(K_Z + \lambda I)^{-1}$ costs $O(b^3)$ where $b$ is the batch size, or $O(b^{2.376})$ using more efficient inverse algorithms like Coppersmith and Winograd (1987). We use the inverse approach with $O(b^3)$ computational cost, which will not be an issue for our method. This is because we consider a mini-batch training to constrain the size of $b$, and the inverse $(K_Z + \lambda I)^{-1}$ does not contain gradients. The computational bottlenecks are gradients computation and their updates.

***Step IV - Converting Existing Contrastive Learning Objectives.*** We short hand $K_{XY}(K_Z + \lambda \mathbf{I})^{-1} K_Z$ as $K_{X \perp\!\!\!\perp Y | Z}$, following notation from prior work (Fukumizu et al., 2007) that short-hands $P_{X|Z} P_{Y|Z}$ as $P_{X \perp\!\!\!\perp Y | Z}$. Now, we plug in the estimation of $e^{f(x,y)}$ using Proposition 2.2 into WeaklySup-InfoNCE (equation 3), Fair-InfoNCE (equation 4), and HardNeg-InfoNCE (equation 5) objectives, coverting them into *Conditional Contrastive learning with Kernel (CCL-K)* objectives:

$$\text{WeaklySup}_{\text{CCLK}} := \mathbb{E}_{\{(x_i, y_i, z_i)\}_{i=1}^b \sim P_{XYZ}^{\otimes b}} \left[ \log \frac{[K_{X \perp\!\!\!\perp Y | Z}]_{ii}}{[K_{X \perp\!\!\!\perp Y | Z}]_{ii} + \sum_{j \neq i} [K_{XY}]_{ij}} \right]. \tag{7}$$

$$\text{Fair}_{\text{CCLK}} := \mathbb{E}_{\{(x_i, y_i, z_i)\}_{i=1}^b \sim P_{XYZ}^{\otimes b}} \left[ \log \frac{[K_{XY}]_{ii}}{[K_{XY}]_{ii} + (b-1)[K_{X \perp\!\!\!\perp Y | Z}]_{ii}} \right]. \tag{8}$$

$$\text{HardNeg}_{\text{CCLK}} := \mathbb{E}_{\{(x_i, y_i, z_i)\}_{i=1}^b \sim P_{XYZ}^{\otimes b}} \left[ \log \frac{[K_{XY}]_{ii}}{[K_{XY}]_{ii} + (b-1)[K_{X \perp\!\!\!\perp Y | Z}]_{ii}} \right]. \tag{9}$$

## 3 RELATED WORK

The majority of the literature on contrastive learning focuses on self-supervised learning tasks (Oord et al., 2018), which leverages unlabeled samples for pretraining representations and then uses the learned representations for downstream tasks. Its applications span various domains, including computer vision (Hjelm et al., 2018), natural language processing (Kong et al., 2019), speech processing (Baevski et al., 2020), and even interdisciplinary (vision and language) across domains (Radford et al., 2021). Besides the empirical success, Arora et al. (2019); Lee et al. (2020); Tsai et al. (2021d) provide theoretical guarantees, showing that contrastively learning can reduce the sample complexity on downstream tasks. The standard self-supervised contrastive frameworks consider the objective that requires only data's pairing information: it learns similar representations between different views of a data (augmented variants of the same image (Chen et al., 2020) or an image-caption pair (Radford et al., 2021)) and dissimilar representations between two random data. We refer to these frameworks as *unconditional* contrastive learning, in contrast to our paper's focus - *conditional* contrastive learning, which considers contrastive objectives that take additional conditioning variables into account. Such conditioning variables can be sensitive information from data (Tsai et al., 2021c), auxiliary information from data (Tsai et al., 2021a), downstream labels (Khosla et al., 2020), or data's embedded features (Robinson et al., 2020; Wu et al., 2020; Kalantidis et al., 2020). It is worth noting that, with additional conditioning variables, the conditional contrastive frameworks extend the self-supervised learning settings to the weakly supervised learning (Tsai et al., 2021a) or the supervised learning setting (Khosla et al., 2020).

Our paper also relates to the literature on few-shot conditional generation (Sinha et al., 2021), which aims to model the conditional generative probability (generating instances according to a conditioning variable) given only a limited amount of paired data (paired between an instance and its corresponding conditioning variable). Its applications span conditional mutual information estimation (Mondal et al., 2020), noisy signals recovery (Candes et al., 2006), image manipulation (Park et al., 2020; Sinha et al., 2021), etc. These applications require generating authentic data, which is notoriously challenging (Goodfellow et al., 2014; Arjovsky et al., 2017). On the contrary, our method models the conditional generative probability via Kernel Conditional Embedding Operator (Song et al., 2013), which generates kernel embeddings but not raw data. Ton et al. (2021) relates to our work and also uses conditional mean embedding to perform estimation regarding the conditional distribution. The differences is that Ton et al. (2021) tries to improve conditional density estimation while this paper aims to resolve the challenge of insufficient samples of the conditional variable. Also, both Ton et al. (2021) and this work consider noise contrastive method, more specifically, Ton et al. (2021) discusses noise contrastive estimation (NCE) (Gutmann and Hyvärinen, 2010), while this work discusses InfoNCE (Oord et al., 2018) objective which is inspired from NCE.

Our proposed method can also connect to domain generalization (Blanchard et al., 2017), if we treat each $z_i$ as a domain or a task indicator (Tsai et al., 2021c). In specific, Tsai et al. (2021c) considers a conditional contrastive learning setup, and one task of it performs contrastive learning from data

across multiple domains, by conditioning on domain indicators to reduce domain-specific information for better generalization. This paper further extends this idea from Tsai et al. (2021c), by using conditional mean embedding to address the challenge of insufficient data in certain conditioning variables (in this case, domains).

## 4 EXPERIMENTS

We conduct experiments on various conditional contrastive learning frameworks that are discussed in Section 2.2: Section 4.1 for the weakly supervised contrastive learning, Section 4.2 for the fair contrastive learning; and Section 4.3 for the hard-negatives contrastive learning.

**Experimental Protocal.** We consider the setup from the recent contrastive learning literature (Chen et al., 2020; Robinson et al., 2020; Wu et al., 2020), which contains stages of pre-training, fine-tuning, and evaluation. In the pre-training stage, on data's training split, we update the parameters in the feature encoder (i.e., $g_{\theta.}(\cdot)$s in equation 2) using the contrastive learning objectives $\big($e.g., InfoNCE (equation 1), WeaklySup$_{\text{InfoNCE}}$ (equation 3), or WeaklySup$_{\text{CCLK}}$ (equation 7)$\big)$. In the fine-tuning stage, we fix the parameters of the feature encoder and add a small fine-tuning network on top of it. On the data's training split, we fine-tune this small network with the downstream labels. In the evaluation stage, we evaluate the fine-tuned representations on the data's test split. We adopt ResNet-50 He et al. (2016) or LeNet-5 LeCun et al. (1998) as the feature encoder and a linear layer as the fine-tuning network. All experiments are performed using LARS optimizer You et al. (2017). More details can be found in Appendix and our released code.

### 4.1 WEAKLY SUPERVISED CONTRASTIVE LEARNING

In this subsection, we perform experiments within the weakly supervised contrastive learning framework (Tsai et al., 2021a), which considers auxiliary information as the conditioning variable $Z$. It aims to learn similar representations for a pair of data with similar outcomes from the conditioning variable (i.e., similar auxiliary information), and vice versa.

**Datasets and Metrics.** We consider three visual datasets in this set of experiments. Data $X$ and $Y$ represent images after applying arbitrary image augmentations. **1) UT-Zappos** (Yu and Grauman, 2014): It contains $50,025$ shoe images over 21 shoe categories. Each image is annotated with 7 binomially-distributed attributes as auxiliary information, and we convert them into 126 binary attributes (Bernoulli-distributed). **2) CUB** (Wah et al., 2011): It contains $11,788$ bird images spanning 200 fine-grain bird species, meanwhile 312 binary attributes are attached to each image. **3) ImageNet-100** (Russakovsky et al., 2015): It is a subset of ImageNet-1k Russakovsky et al. (2015) dataset, containing 0.12 million images spanning 100 categories. This dataset does not come with auxiliary information, and hence we extract the 512-dim. visual features from the CLIP (Radford et al., 2021) model (a large pre-trained visual model with natural language supervision) to be its auxiliary information. Note that we consider different types of auxiliary information. For **UT-Zappos** and **CUB**, we consider discrete and human-annotated attributes. For **ImageNet-100**, we consider continuous and pre-trained language-enriched features. We report the top-1 accuracy as the metric for the downstream classification task.

**Methodology.** We consider the WeaklySup$_{\text{CCLK}}$ objective (equation 7) as the main method. For WeaklySup$_{\text{CCLK}}$, we perform the sampling process $\{(x_i, y_i, z_i)\}_{i=1}^{b} \sim P_{XYZ}^{\otimes b}$ by first sampling an image $\text{im}_i$ along with its auxiliary information $z_i$ and then applying different image augmentations on $\text{im}_i$ to obtain $(x_i, y_i)$. We also study different types of kernels for $K_Z$. On the other hand, we select two baseline methods. The first one is the unconditional contrastive learning method: the InfoNCE objective (Oord et al., 2018; Chen et al., 2020) (equation 1). The second one is the conditional contrastive learning baseline: the WeaklySup$_{\text{InfoNCE}}$ objective (equation 3). The difference between WeaklySup$_{\text{CCLK}}$ and WeaklySup$_{\text{InfoNCE}}$ is that the latter requires sampling a pair of data with the same outcome from the conditioning variable $\big($i.e., $(x, y) \sim P_{X|Z=z} P_{Y|Z=z}\big)$. However, as suggested in Section 2.3, directly performing conditional sampling is challenging if there is not enough data to support the conditioning sampling procedure. Such limitation exists in our datasets: **CUB** has on average 1.001 data points per $Z$'s configuration, and **ImageNet-100** has only 1 data point per $Z$'s configuration since its conditioning variable is continuous and each instance from the dataset has a unique $Z$. To avoid this limitation, in WeaklySup$_{\text{InfoNCE}}$ clusters the data to ensure that data within the same cluster are abundant and have similar auxiliary information, and then treating

| | UT-Zappos | CUB | ImageNet-100 |
|---|---|---|---|
| *Unconditional Contrastive Learning Methods* | | | |
| InfoNCE | $77.8 \pm 1.5$ | $14.1 \pm 0.7$ | $76.2 \pm 0.3$ |
| *Conditional Contrastive Learning Methods* | | | |
| WeaklySup$_{\text{InfoNCE}}$ | $84.6 \pm 0.4$ | $20.6 \pm 0.5$ | $81.4 \pm 0.4$ |
| WeaklySup$_{\text{CCLK}}$ (ours) | $\mathbf{86.6} \pm 0.7$ | $\mathbf{29.9} \pm 0.3$ | $\mathbf{82.4} \pm 0.5$ |

| Kernels | UT-Zappos | CUB | ImageNet-100 |
|---|---|---|---|
| RBF | $86.5 \pm 0.5$ | $32.3 \pm 0.5$ | $81.8 \pm 0.4$ |
| Laplacian | $86.8 \pm 0.3$ | $32.1 \pm 0.5$ | $80.2 \pm 0.3$ |
| Linear | $86.5 \pm 0.4$ | $29.4 \pm 0.8$ | $77.5 \pm 0.3$ |
| Cosine | $86.6 \pm 0.7$ | $29.9 \pm 0.3$ | $82.4 \pm 0.5$ |

Table 2: Object classification accuracy (%) under the weakly supervised contrastive learning setup. Left: results of the proposed method and the baselines. Right: different types of kernel choice in WeaklySup$_{\text{CCLK}}$.

the clustering information as the new conditioning variable. The result of WeaklySup$_{\text{InfoNCE}}$ is reported by selecting the optimal number of clusters via cross-validation.

**Results.** We show the results in Table 2. First, WeaklySup$_{\text{CCLK}}$ shows consistent improvements over unconditional baseline InfoNCE, with absolute improvements of 8.8%, 15.8%, and 6.2% on UT-Zappos, CUB and, ImageNet-100 respectively. This is because the conditional method utilizes the additional auxiliary information (Tsai et al., 2021a). Second, WeaklySup$_{\text{CCLK}}$ performs better than WeaklySup$_{\text{InfoNCE}}$, with absolute improvements of 2%, 9.3%, and 1%. We attribute better performance of WeaklySup$_{\text{CCLK}}$ over WeaklySup$_{\text{InfoNCE}}$ to the following fact: WeaklySup$_{\text{InfoNCE}}$ first performs clustering on the auxiliary information and considers the new clusters as the conditioning variable, while WeaklySup$_{\text{CCLK}}$ directly considers the auxiliary information as the conditioning variable. The clustering in WeaklySup$_{\text{InfoNCE}}$ may lose precision of the auxiliary information and may negatively affect the quality of the auxiliary information incorporated in the representation. Ablation study on the choice of kernels has shown the consistent performance of WeaklySup$_{\text{CCLK}}$ across different kernels on the UT-Zappos, CUB and ImageNet-100 dataset, where we consider the following kernel functions: RBF, Laplacian, Linear and Cosine. Most kernels have similar performances, except that linear kernel is worse than others on ImageNet-100 (by at least 2.7%).

## 4.2 FAIR CONTRASTIVE LEARNING

In this subsection, we perform experiments within the fair contrastive learning framework (Tsai et al., 2021c), which considers sensitive information as the conditioning variable $Z$. It fixes the outcome of the sensitive variable for both the positively-paired and negatively-paired samples in the contrastive learning process, which leads the representations to ignore the effect from the sensitive variable.

**Datasets and Metrics.** Our experiments focus on continuous sensitive information, to echo with the limitation of the conditional sampling procedure in Section 2.3. Nonetheless, existing datasets mostly consider discrete sensitive variables, such as gender or race. Therefore, we synthetically create **ColorMNIST** dataset, which randomly assigns a continuous RBG color value for the background in each handwritten digit image in the MNIST dataset (LeCun et al., 1998). We consider the background color as sensitive information. For statistics, **ColorMNIST** has $60,000$ colored digit images across 10 digit labels. Similar to Section 4.1, data $X$ and $Y$ represent images after applying arbitrary image augmentations. Our goal is two-fold: we want to see how well the learned representations 1) perform on the downstream classification task and 2) ignore the effect from the sensitive variable. For the former one, we report the top-1 accuracy as the metric; for the latter one, we report the Mean Square Error (MSE) when trying to predict the color information. Note that the MSE score is higher the better since we would like the learned representations to contain less color information.

**Methodology.** We consider the Fair$_{\text{CCLK}}$ objective (equation 8) as the main method. For Fair$_{\text{CCLK}}$, we perform the sampling process $\{(x_i, y_i, z_i)\}_{i=1}^{b} \sim P_{XYZ}^{\otimes b}$ by first sampling a digit image $\text{im}_i$ along with its sensitive information $z_i$ and then applying different image augmentations on $\text{im}_i$ to obtain $(x_i, y_i)$. We select the unconditional contrastive learning method - the InfoNCE objective (equation 1) as our baseline. We also consider the Fair$_{\text{InfoNCE}}$ objective (equation 4) as a baseline, by clustering the continuous conditioning variable $Z$ into one of the following: 3, 5, 10, 15, or 20 clusters using K-means.

| | Top-1 Accuracy ($\uparrow$) | MSE ($\uparrow$) |
|---|---|---|
| *Unconditional Contrastive Learning Methods* | | |
| InfoNCE | $84.1 \pm 1.8$ | $48.8 \pm 4.5$ |
| *Conditional Contrastive Learning Methods* | | |
| Fair$_{\text{InfoNCE}}$ | $\mathbf{85.9} \pm 0.4$ | $\mathbf{64.9} \pm 5.1$ |
| Fair$_{\text{CCLK}}$ (ours) | $\mathbf{86.4} \pm 0.9$ | $64.7 \pm 3.9$ |

Table 3: Classification accuracy (%) under the fair contrastive learning setup, and the MSE (higher the better) between color in an image and color prediction by the image's representation. A higher MSE indicates less color information from the original image is contained in the learned representation.

**Results.** We show the results in Table 3. Fair$_{\text{CCLK}}$ is consistently better than the InfoNCE, where the absolute accuracy improvement is 2.3% and the relative improvement of MSE (higher the better) is 32.6% over InfoNCE. We report the result of using the Cosine kernel and provide the ablation study of different kernel choices in the Appendix. This result suggests that the proposed Fair$_{\text{CCLK}}$ can

achieve better downstream classification accuracy while ignoring more sensitive information (color information) compared to the unconditional baseline, suggesting that our method can achieve a better level of fairness (by excluding color bias) without sacrificing performance. Next we compare to $\text{Fair}_{\text{InfoNCE}}$ baseline, and we report the result using the 10-cluster partition of $Z$ as it achieves the best top-1 accuracy. Compared to $\text{Fair}_{\text{InfoNCE}}$, $\text{Fair}_{\text{CCLK}}$ is better in downstream accuracy, while slightly worse in MSE (a difference of 0.2). For $\text{Fair}_{\text{InfoNCE}}$, if the number of discretized value of $Z$ increases, the MSE in general grows, but the accuracy peaks at 10 clusters and then declines. This suggests that $\text{Fair}_{\text{InfoNCE}}$ can remove more sensitive information as the granularity of $Z$ increases, but may hurt the downstream task performance. Overall, the $\text{Fair}_{\text{CCLK}}$ performs slightly better than $\text{Fair}_{\text{InfoNCE}}$, and do not need clustering to discretize $Z$.

### 4.3 HARD-NEGATIVES CONTRASTIVE LEARNING

In this subsection, we perform experiments within the hard negative contrastive learning framework (Robinson et al., 2020), which considers the embedded features as the conditioning variable $Z$. Different from conventional contrastive learning methods (Chen et al., 2020; He et al., 2020) that considers learning dissimilar representations for a pair of random data, the hard negative contrastive learning methods learn dissimilar representations for a pair of random data when they have similar embedded features (i.e., similar outcomes from the conditioning variable).

**Datasets and Metrics.** We consider two visual datasets in this set of experiments. Data $X$ and $Y$ represent images after applying arbitrary image augmentations. **1) CIFAR10** (Krizhevsky et al., 2009): It contains $60,000$ images spanning 10 classes, e.g. automobile, plane, or dog. **2) ImageNet-100** (Russakovsky et al., 2015): It is the same dataset that we used in Section 2. We report the top-1 accuracy as the metric for the downstream classification task.

**Methodology.** We consider the $\text{HardNeg}_{\text{CCLK}}$ objective (equation 9) as the main method. For $\text{HardNeg}_{\text{CCLK}}$, we perform the sampling process $\{(x_i, y_i, z_i)\}_{i=1}^{b} \sim P_{XYZ}^{\otimes b}$ by first sampling an image $\text{im}_i$, then applying different image augmentations on $\text{im}_i$ to obtain $(x_i, y_i)$, and last defining $z_i = g_{\theta_X}(x_i)$. We select two baseline methods. The first one is the unconditional contrastive learning method: the InfoNCE objective (Oord et al., 2018; Chen et al., 2020) (equation 1). The second one is the conditional contrastive learning baseline: the $\text{HardNeg}_{\text{InfoNCE}}$, objective (Robinson et al., 2020), and we report its result directly using the released code from the author (Robinson et al., 2020).

**Results.** From Table 4, first, $\text{HardNeg}_{\text{CCLK}}$ consistently shows better performances over the InfoNCE baseline, with absolute improvements of $1.8\%$ and $3.1\%$ on CIFAR-10 and ImageNet-100 respectively. This suggests that the hard negative sampling effectively improves the downstream performance, which is in accordance with the observation by Robinson et al. (2020). Next, $\text{HardNeg}_{\text{CCLK}}$ also performs better than the $\text{HardNeg}_{\text{InfoNCE}}$ baseline, with absolute improvements of $0.3\%$ and $2.0\%$ on CIFAR-10 and ImageNet-100 re-

|  | CIFAR-10 | ImageNet-100 |
|---|---|---|
| *Unconditional Contrastive Learning Methods* | | |
| InfoNCE | $89.9 \pm 0.2$ | $77.8 \pm 0.4$ |
| *Conditional Contrastive Learning Methods* | | |
| $\text{HardNeg}_{\text{InfoNCE}}$ | $91.4 \pm 0.2$ | $79.2 \pm 0.5$ |
| $\text{HardNeg}_{\text{CCLK}}$ (ours) | $\mathbf{91.7 \pm 0.1}$ | $\mathbf{81.2 \pm 0.2}$ |

Table 4: Classification accuracy (%) under the hard negatives contrastive learning setup.

spectively. Both methods construct hard negatives by assigning a higher weight to a random paired data that are close in the embedding space and a lower weight to a random paired data that are far in the embedding space. The implementation by Robinson et al. (2020) uses Euclidean distances to measure the similarity and $\text{HardNeg}_{\text{CCLK}}$ uses the smoothed kernel similarity (i.e., $(K_Z + \lambda \mathbf{I})^{-1} K_Z$ in Proposition 2.2) to measure the similarity. Empirically our approach performs better.

## 5 CONCLUSION

In this paper, we present CCL-K, the **C**onditional **C**ontrastive **L**earning objectives with **K**ernel expressions. CCL-K avoids the need to perform explicit conditional sampling in conditional contrastive learning frameworks, alleviating the insufficient data problem of the conditioning variable. CCL-K uses the Kernel Conditional Embedding Operator, which first defines the kernel similarities between the conditioning variable's values and then samples data that have similar values of the conditioning variable. CCL-K is can directly work with continuous conditioning variables, while prior work requires binning or clustering to ensure sufficient data for each bin or cluster. Empirically, CCL-K also outperforms conditional contrastive baselines tailored for weakly-supervised contrastive learning, fair contrastive learning, and hard negatives contrastive learning. An interesting future direction is to add more flexibility to CCL-K, by relaxing the kernel similarity to arbitrary similarity measurements.

## 6 Ethics Statement

Because our method can improve removing sensitive information in contrastive learning representations, our contribution can have a positive impact on fairness and privacy, where biases or user-specific information should be excluded from the representation. However, the conditioning variable must be predefined, so our method cannot directly remove any biases that are implicit and are not captured by a variable in the dataset.

## 7 Reproducibility Statement

We provide an anonymous source code link for reproducing our result in the supplementary material and include complete files which can reproduce the data processing steps for each dataset we use in the supplementary material.

## Acknowledgements

The authors would like to thank the anonymous reviewers for helpful comments and suggestions. This work is partially supported by the National Science Foundation IIS1763562, IARPA D17PC00340, ONR Grant N000141812861, Facebook PhD Fellowship, BMW, National Science Foundation awards 1722822 and 1750439, and National Institutes of Health awards R01MH125740, R01MH096951 and U01MH116925. KZ would like to acknowledge the support by the National Institutes of Health (NIH) under Contract R01HL159805, by the NSF-Convergence Accelerator Track-D award 2134901, and by the United States Air Force under Contract No. FA8650-17-C7715. HZ would like to thank support from a Facebook research award. Any opinions, findings, conclusions, or recommendations expressed in this material are those of the author(s) and do not necessarily reflect the views of the sponsors, and no official endorsement should be inferred.

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

## A    CODE

We include our code in a Github link: [https://github.com/Crazy-Jack/CCLK-release](https://github.com/Crazy-Jack/CCLK-release)

## B    ABLATION OF DIFFERENT CHOICES OF KERNELS

We report the kernel choice ablation study for $\text{Fair}_{\text{CCLK}}$ and $\text{HardNeg}_{\text{CCLK}}$ (Table 5). For the $\text{Fair}_{\text{CCLK}}$, we consider the synthetically created ColorMNIST dataset, where the background color for each image digit image is randomly assigned. We report the accuracy of the downstream classification task, as well as the Mean Square Error (MSE) between the assigned background color and a color predicted by the learned representation. A higher MSE score is better because we would like the learned representations to contain less color information. In Table 5, we consider the following kernel functions: RBF, Polynomial (degree of 3), Laplacian and Cosine. The performances using different kernels are consistent.

For the $\text{HardNeg}_{\text{CCLK}}$, we consider the CIFAR-10 dataset for the ablation study and use the top-1 accuracy for object classification as the metric. In Table 5, we consider the following kernel functions: Linear, RBF, and Polynomial (degree of 3), Laplacian and Cosine. The performances using different kernels are consistent.

Lastly, we also provide the ablation of different $\sigma^2$ when using the RBF kernel. First, the performance trend does change too much and is sensitive to different bandwidths for the RBF kernel. In Table 7 we show the result of using different $\sigma^2$ in the RBF kernel. We found that using $\sigma^2 = 1$ significantly hurts performance (only 3.0%), using $\sigma^2 = 1000$ is also sub-optimal. The performances of $\sigma^2 = 10, 100, 500$ are close.

## C    ADDITIONAL RESULTS ON CIFAR10

In Section 4.3 in the main text, we report the results of $\text{HardNeg}_{\text{CCLK}}$ as well as baseline methods on CIFAR10 dataset by training with 400 epochs using 256 batch size. Here we also include the results with larger batch size (512 batch size) and longer training time (1000 epochs). The results are summarized in Table 6. For reference, we name the training procedure with 256 batch size and 400 epochs **setting 1** and the one with 512 batch size and 1000 epochs as **setting 2**. From Table 6 we observe that $\text{HardNeg}_{\text{CCLK}}$ still has better performance comparing to $\text{HardNeg}_{\text{InfoNCE}}$, suggesting that our method is solid. However, the performance differences shrink between the vallina InfoNCE method and the Hard Negative mining approaches in general because the benefit of hard negative mining mainly lies in the training efficiency, i.e., spend less training iteration on discriminating the negative samples that have been already pushed away.

## D    FAIR$_{\text{InfoNCE}}$ ON COLORMNIST

Here we include the results of performing $\text{Fair}_{\text{InfoNCE}}$ on the ColorMNIST dataset as a baseline. We implemented $\text{Fair}_{\text{InfoNCE}}$ based on Tsai et al. (2021c), where the idea is to remove the information of the conditional variable by sampling positive and negative pairs from the same outcome of the conditional variable at each iteration. Same as our previous setup in Section 4.2, we evaluate the results by the accuracy of the downstream classification task as well as the MSE value, both of which are deemed higher as the better (as we want the learned representation to contain less color information, so a larger MSE is desirable because it means the representation contains less color information for reconstruction.) To perform $\text{Fair}_{\text{InfoNCE}}$ and condition on the continuous color information, we need to discretize the color variable using clustering method such as K-means. We use K-means to cluster samples based on their color variable, and we use the following numbers of clusters: $\{3, 5, 10, 15, 20\}$.

As shown in Table 8, with the number of clusters increasing, the downstream accuracy first increases then drops, peaking at number of clusters being 10. The MSE values continue increase as number of clusters increase. This suggests that $\text{Fair}_{\text{InfoNCE}}$ can remove more sensitive information as the granularity of $Z$ increases, but the downstream task performance may decrease.

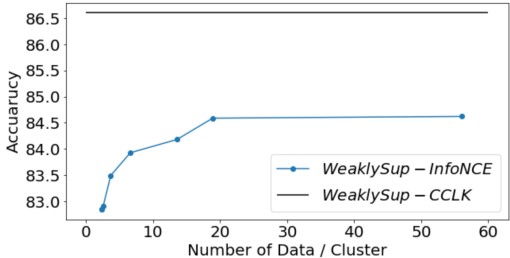

Figure 3: Illustration of the problem of insufficient samples in conditional contrastive learning. When the average number of samples per outcome (cluster) of the conditional variable is small (towards the left of the x-axis in the figure), the previous conditional contrastive learning framework WeaklySup$_{\text{InfoNCE}}$ (blue) suffers, while the proposed WeaklySup$_{\text{CCLK}}$ (black) outperforms WeaklySup$_{\text{InfoNCE}}$ significantly and is very stable regardless of whether the samples are sufficient (towards the right of the x-axis) or insufficient (towards the left of the x-axis).

## E  PERFORMANCE UNDER INSUFFICIENT NUMBER OF SAMPLES

We would illustrate why the insufficient sample would be a problem for conditional contrastive learning. We provide comparison of performances under different number of conditioning data. To be specific, we provide Figure 3, where the x-axis is the averaged number of data samples per discretized conditioning variable (cluster) for conditional contrastive learning, if we use framework like the WeaklySup$_{\text{InfoNCE}}$. The dataset is UT-Zappos and the conditioning variable is the annotative attributes. The discretization is done by grouping instances that share the same annotative attributes to the same cluster. The blue line is WeaklySup$_{\text{InfoNCE}}$, which requires discretized conditioning variables. The black line represents the proposed WeaklySup$_{\text{CCLK}}$ which does not require discretization. As we can see from the figure, the performance of WeaklySup$_{\text{InfoNCE}}$ suffers when the number of data per cluster (conditioning variable) is small, and WeaklySup$_{\text{CCLK}}$ outperforms WeaklySup$_{\text{InfoNCE}}$ in all cases. From this example, we can see that when the data is very insufficient (towards the origin in this figure), the proposed WeaklySup$_{\text{CCLK}}$ outperforms WeaklySup$_{\text{InfoNCE}}$ significantly.

## F  DATASET DETAILS

We provide the training details of experiments conducted on the following datasets: UT-Zappos50 (Yu and Grauman, 2014), CUB-200-2011 (Wah et al., 2011), CIFAR-10 (Krizhevsky et al., 2009), ColorMNIST (our creation), and ImageNet-100 (Russakovsky et al., 2015).

### F.1  UT-ZAPPOS50K

The following section describes the experiments we performed on UT-Zappos50K dataset.

**Accessiblity**   The dataset is attributed to (Yu and Grauman, 2014) and available at the link: http://vision.cs.utexas.edu/projects/finegrained/utzap50k. The dataset is for non-commercial use only.

| Fair$_{\text{CCLK}}$ | Top-1 Accuracy (↑) | MSE (↑) |
|---|---|---|
| RBF kernel | $86.2 \pm 0.5$ | $57.6 \pm 10.6$ |
| Polynomial kernel | $86.7 \pm 0.5$ | $61.3 \pm 9.4$ |
| Laplacian kernel | $85.0 \pm 0.9$ | $72.8 \pm 13.2$ |
| Cosine kernel | $86.4 \pm 0.9$ | $64.7 \pm 3.9$ |

| HardNeg$_{\text{CCLK}}$ | CIFAR-10 |
|---|---|
| Linear kernel | $91.5 \pm 0.2$ |
| RBF kernel | $91.7 \pm 0.1$ |
| Polynomial kernel | $90.3 \pm 0.4$ |

Table 5: Ablation study of different types of kernel choices. Left: digit classification accuracy of Fair$_{\text{CCLK}}$ in ColorMNIST, and MSE (higher the better) between the color in the original image and the color predicted based on the learned representation from that image. Higher MSE is better because we intend to remove color information in the representation. Right: classification accuracy of HardNeg$_{\text{CCLK}}$ on CIFAR-10 object classification. The performances using different kernels in both settings are consistent.

|  | CIFAR-10 (setting 1) | CIFAR-10 (setting 2) |
|---|---|---|
| *Unconditional Contrastive Learning Methods* | | |
| InfoNCE | $89.9 \pm 0.2$ | $93.4 \pm 0.1$ |
| *Conditional Contrastive Learning Methods* | | |
| HardNeg$_{\text{InfoNCE}}$ | $91.4 \pm 0.2$ | $93.6 \pm 0.2$ |
| HardNeg$_{\text{CCLK}}$ (ours) | $\textbf{91.7} \pm 0.1$ | $\textbf{93.9} \pm 0.1$ |

Table 6: Results of HardNeg$_{\text{CCLK}}$ on CIFAR10 dataset with two different training settings.

| RBF $\sigma^2$ | **1** | **10** | **100** | **500** | **1000** |
|---|---|---|---|---|---|
| **Accuracy (%)** | $3.0 \pm 1.5$ | $30.9 \pm 0.3$ | $32.2 \pm 0.4$ | $32.0 \pm 0.2$ | $24.4 \pm 0.9$ |

Table 7: Result of WeaklySup$_{\text{CCLK}}$ under different hyper-parameters $\sigma^2$ using the RBF kernel in the CUB dataset.

| Number of Clusters | **Top-1 Accuracy ($\uparrow$)** | **MSE ($\uparrow$)** |
|---|---|---|
| 3 | $82.12 \pm 0.3$ | $56.27 \pm 4.9$ |
| 5 | $84.55 \pm 0.4$ | $58.67 \pm 4.8$ |
| 10 | $85.90 \pm 0.4$ | $64.91 \pm 5.1$ |
| 15 | $85.22 \pm 0.4$ | $65.02 \pm 5.0$ |
| 20 | $84.23 \pm 0.3$ | $65.11 \pm 4.9$ |

Table 8: Results of Fair$_{\text{InfoNCE}}$ on the colored MNIST dataset with different numbers of clusters.

**Data Processing**   The dataset contains images of shoe from Zappos.com. We downsamples the images to $32 \times 32$. The official dataset has 4 large categories following 21 sub-categories. We utilize 21 subcategories for all our classification tasks. The dataset comes with 7 attributes as auxiliary information. We binarize the 7 discrete attributes into 126 binary attributes. We consider our conditional variable Z is this 128 dimensional variable.

*Training and Test Split*: We randomly split train-validation images by $7 : 3$ ratio, resulting in $35,017$ train data and $15,008$ validation dataset.

**Network Design and Optimization**   We use ResNet-50 architecture to serve as a backbone for the encoder. To compensate the 32x32 image size, we change the first 7x7 2D convolution to 3x3 2D convolution and remove the first max pooling layer in the normal ResNet-50 (See code for details). This allows a finer grain of information processing. After using the modified ResNet-50 as the encoder, we include a 2048-2048-128 Multi-Layer Perceptron (MLP) as the projection head. Batch normalization is used after each 2048 activation layers. During the evaluation, we discard the projection head and train a linear layer on top of the encoder's output. We train 1000 epochs for all experiments with LARS optimizer (base learning rate 1.5 and scale learning rate based on our batch size divided by 256) with batch size 152 on 4 NVIDIA 1080ti GPUs. It takes about 16 hours to finish 1000 epochs training.

## F.2   CUB-200-2011

The following section describes the experiments we performed on CUB-200-2011 dataset.

**Accessiblity**   CUB-200-2011 is created by Wah et al. (2011) and is a fine-grained dataset for bird species. It can be downloaded from the link: http://www.vision.caltech.edu/visipedia/CUB-200-2011.html. The usage is restricted to noncommercial research and educational purposes.

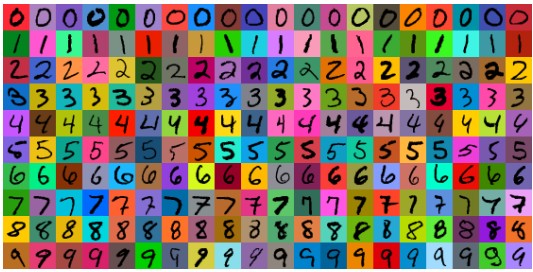

Figure 4: Creation of ColorMNIST for experiments on Fair-InfoNCE validation.

**Data Processing**    The original dataset contains 200 birds categories over $11,788$ images with 312 binary attributes attached to each image. The image is rescaled to $224 \times 224$.

*Train Test Split*: We follow the original train-validation split, resulting in $5,994$ train images and $5,794$ validation images. We combine the original training and validation set as our training set and use the original test set as our validation set. The resulting training set contains $6,871$ images and the validation set contains $6,918$ images.

**Network Design and Optimization**    We use ResNet-50 architecture as an encoder. We choose 2048-2048-128 MLP as the projection head. Batch normalization is used after each 2048 activation layers.Different than UT-Zappos dataset, we directly employ the original design of ResNet-50 since we are training on 224x224 images. Similarly, LARS is used for optimization during the contrastive learning pretraining and we fine tune a linear layer and use Limited-memory BFGS (L-BFGS (Liu and Nocedal, 1989)) optimizer after pretraining. All experiments are run with 1000 pretraining iterations and 500 L-BFGS fine tuning steps. We use 128 batch size and train it on 4 1080ti NVIDIA GPUs. It takes about 13 hours to finish 1000 epochs training.

## F.3   CIFAR-10

The following section describes the experiments we performed on CIFAR-10.

**Accessibility**    CIFAR-10 (Krizhevsky et al., 2009) is an object detection dataset with $60,000$ $32 \times 32$ images in 10 classes. The test set includes $10,000$ images. The dataset can be downloaded at https://www.cs.toronto.edu/~kriz/cifar.html.

**Data Processing and Train and Test split**    We use the training and test split from the original dataset.

**Network Design and Optimization**    We employ ResNet-50 backbone architecture, but we change the first 7x7 2D convolution to 3x3 2D convolution and remove the first max pooling layer in the normal ResNet-50 (See code for details). This allows better results on CIFAR10 as this dataset consists of 32x32 resolution images. 2048-2048-128 projection head is employed during contrastive learning. Batch normalization is used after each 2048 activation layers. There are two CIFAR10 training settings we consider. The first setting, which is reported in the main text, trains contrastive learning with 256 batch size for 400 epochs. It takes a 4 GPU 1080ti Machine 8 hours to finish the pretraining. For the second setting where we train with 512 batch size for 1000 epochs, it takes an DGX-1 machine 48 hours to finish training. We use LARS optimizer for all CCL-K related experiments with base lr=1.5 and base batch size equals 256.

## F.4   CREATION OF COLORMNIST

**Accessiblity**    We create ColorMNIST dataset for experiments in Section 4.2 in the main text. The train and test split images can be accessed from our anonymous Github link (Section A). We allow any non-commerical usage of our dataset.

**Data Processing**    As discussed in Section 4.2 in the main text, we create the **ColorMNIST** dataset by assigning a random sampled color as MNIST's background. Images are converted into 32x32 resolution and only the background is augmented with the sampled color while the digit stroke pixel remains black. Examples of the **ColorMNIST** images are shown in Figure 4.

*Training and Test Split* We follow the original MNIST train/test split, resulting in 60,000 training images and 10,000 testing images spanning 10 digit categories.

**Network Design and Optimization**    To train our model, we use LeNet-5 (LeCun et al., 1998) as backbone architecture and use 2 layer linear projection head to project it to 128 dimension. We use LARS (You et al., 2017) as our optimizer. After our network is pretrained using contrastive learning, we discard the head annd fine tune a linear layer use Limited-memory BFGS (L-BFGS (Liu and Nocedal, 1989)) as optimizer. All experiments are run with 1175 pretraining iterations and 500 L-BFGS fine tuning steps.

### F.5    IMAGENET-100

The following section describes the experiments we performed on ImageNet-100 dataset in Section 4 in the main text.

**Accessibility**    This dataset is a subset of ImageNet-1K dataset, which comes from the ImageNet Large Scale Visual Recognition Challenge (ILSVRC) 2012-2017 (Russakovsky et al., 2015). ILSVRC is for non-commercial research and educational purposes and we refer to the ImageNet official site for more information: `https://www.image-net.org/download.php`.

**Data Processing**    We select 100 classes from ImageNet-1K to conduct experiments. Selected class names can be accessed from our Github link.

*Training and Test Split*: The training split contains $128,783$ images and the test split contains $5,000$ images. The images are rescaled to size $224 \times 224$.

**Network Design and Optimization Hyper-parameters**    We use conventional ResNet-50 as the backbone for the encoder. 2048-2048-128 MLP layer and a $l2$ normalization layer is used after the encoder during training and discarded in the linear evaluation protocol. Batch normalization is used after each 2048 activation layers. For optimization, we choose 128 batch size for WeaklySup$_{\text{CCLK}}$ setting and 512 batch size for HardNeg$_{\text{CCLK}}$. Open AI CLIP model (Radford et al., 2021) is used to extract continuous feature from the raw image. WeaklySup$_{\text{InfoNCE}}$, we discretize the $Z$ space using Kmeans clustering with k=100, 200, 500, 2500. The best result of WeaklySup$_{\text{InfoNCE}}$ is produced by k=200. All experiments are trained with 200 epochs and require 53 hours of training on DGX machine with 8 Tesla P100 GPUs.

