# OpenReview forum: "Conditional Contrastive Learning with Kernel"
_ICLR.cc/2022/Conference — ICLR 2022 Poster_

### Official Review · Reviewer_F3Tg · 2021-11-02

**Correctness:** 3
**Technical Novelty And Significance:** 3
**Empirical Novelty And Significance:** 3
**Recommendation:** 5
**Confidence:** 4

**Main Review:**

This paper well extends the Kernel Conditional Embedding Operator to the contrastive learning framework. The authors present their method in both clear formulation and extensive experiments in three different tasks. The paper is generally correct but a little bit hard to follow. Some motivations mentioned in the introduction are not well addressed in the method and experiments part (e.g. CCL-K can work with insufficient data, for continuous learning, etc.), thus I am still a little bit confused about the advantage of CCL-K in practice.

From the side of technique in the paper, the proposed sampling method is not novel, as the weighted sum form has been discussed by various paper [1-3]. The novelty lies in the weight estimation, which is achieved by the kernel score.  However, the difference between calculating similarity in the original space and the encoder space is not elaborated enough. The results in Table 2 (right table) seems to show this difference is not significant, which makes the usage of kernel a little bit trivial.


The experiments show the effectiveness of CCL-K compared to InfoNCE loss in weekly-supervised CL, fair CL, but in the hard-negative CL case the improvement is a little bit marginal compared to hard-negative InfoNCE on CIFAR10. Moreover, some papers involving different negative sampling strategy are not compared in the experiments, though they are cited.

[1] Contrastive learning with hard negative samples.
[2] Conditional negative sampling for contrastive learning of visual representations.
[3] Contrastive Attraction and Contrastive Repulsion for Representation Learning.


**Summary Of The Paper:**

This paper presents Conditional Contrastive Learning with Kernel, a kernel sampling method for contrastive learning.  This sampling method leverages the Kernel Conditional Embedding Operator and measures the similarity between the conditioning variables. The authors test the performance of CCL-K in three application scenarios: weakly supervised contrastive learning, fair contrastive learning, and hard negative contrastive learning.

**Summary Of The Review:**

This paper proposes an importance weight sampling technique to improve the InfoNCE loss, where the weight is estimated by using kernel embedding operator, but as the evidence of advantages of using the kernel estimation is not sufficiently shown. I am inclined to recommend a reject of this paper.

---

> ### Author Response · Authors · 2021-11-19
> **Response to Reviewer F3Tg**
>
> We thank the reviewer for the comments. Please see the following response. Please let us know if any of the concerns are not addressed by our response.
>
> **(Contribution and CCLK in Practice)** We like to restate our contribution is resolving the insufficient data problem when performing conditional contrastive representation learning (please refer to general response in this rebuttal). Such insufficient data problem becomes an inevitable problem when the conditioning variable is continuous. To see this, suppose the conditioned variable is continuous body temperature. Then, performing conditional contrastive learning requires us to sample data that have the same body temperature. Yet, in practice, it is hard for us to find different people that have the exact same body temperature, which makes the ordinary conditional sampling approaches infeasible. To the best of our knowledge, our paper is the first to address this research challenge.
>
> **(Remark on Technical Novelty)** The weight estimation is only a portion of our technical novelty and contribution. As discussed in **(Contribution and CCLK in Practice)**, our novelty and contribution is resolving a research challenge in conditional contrastive learning. The core idea of our approach is to avoid direct conditional sampling (direct conditional sampling will be challenging under the insufficient data problem) by having the conditional mean embedding [1] as an approximation. Using the conditional mean embedding [1] allows an interpretation of weighted-summing the data, and how we determine these weights is very different from the weighting scheme in the suggested work [2-4]. To be clear, we determine the weights via similarities between the conditioned values $z_i$, while the suggested works [2-4] determine the weights via similarities between the data. Note that our method is the generalization of the weighting scheme in the suggested works [2-4]. We can specify the conditioned variable as the embedding of data $Z=g_{\theta_X}(x)$ (see our proposed HardNeg_CCLK) to realize [2-4].
>
> In terms of Table 2 (comparison of using different kernels), our objectives (Fair_CCLK/WeakSup_CCLK/HardNeg_CCLK) have a kernel formulation, but it does not mean using non-linear kernels is always better. As shown in Table 2 in the original submission and the new results below, we empirically find selecting the linear kernel is good enough for our objectives in UT-Zappos and CUB datasets, although in ImageNet-100 the linear kernel is slightly off. Nevertheless, we would clarify why we want to use kernels. Our contribution is devising a kernelized approach to resolve the insufficient data problem when performing conditional contrastive representation learning (please refer to the general response in this rebuttal). The core idea of our kernelized approach is to avoid direct conditional sampling (direct conditional sampling will be challenging under the insufficient data problem) by having the conditional mean embedding as an approximation. Using the conditional mean embedding allows a kernel formulation for our objectives (Fair_CCLK/WeakSup_CCLK/HardNeg_CCLK), but it does not mean that we should definitely use non-linear kernels. Allowing a wide range of kernel choices is simply a plus for our methods.
>
> | Kernels      | CUB | ImageNet-100
> | ----------- | ----------- | -----------
> | RBF     | 32.3      | 81.8
> | Laplacian   | 32.1       | 80.2
> | Linear | 29.4  | 77.5
> | Cosine | 29.9 | 82.4
>
> **(Comparisons with Baselines)** Since our submission focuses on addressing the challenge in conditional contrastive learning, we select the representative (SOTA) methods: WeakSup_InfoNCE [5], Fair_InfoNCE [6], and HardNeg_InfoNCE [7] as our baselines.
>
> [1] Kernel embeddings of conditional distributions: A unified kernel framework for nonparametric inference in graphical models. Signal Processing'13.
>
> [2] Contrastive learning with hard negative samples.
>
> [3] Conditional negative sampling for contrastive learning of visual representations.
>
> [4] Contrastive Attraction and Contrastive Repulsion for Representation Learning.
>
> [5] Integrating Auxiliary Information in Self-supervised Learning. arXiv preprint arXiv:2106.02869.
>
> [6] Conditional Contrastive Learning: Removing Undesirable Information in Self-Supervised Representations. arXiv preprint arXiv:2106.02866.
>
> [7] Contrastive learning with hard negative samples. arXiv preprint arXiv:2010.04592.

---

### Official Review · Reviewer_imU8 · 2021-11-02

**Correctness:** 4
**Technical Novelty And Significance:** 3
**Empirical Novelty And Significance:** 2
**Recommendation:** 6
**Confidence:** 4

**Main Review:**

Reasons to accept the paper:
- the paper proposed a novel way to use more contrastive examples by weighting the datapoints using CME.
- the paper is clearly written with a simple idea applied to conditional contrastive learning
- the experimental results look very promising on a variety of experiments

Reasons to reject the paper:
- the paper completely dismisses the work done in [1] which uses almost the exact same idea but in the meta learning setting and conditional density estimation. Also [1] uses NCE instead of batch wise representation learning for negative samples. However, it is undeniable that the objective function proposed in this paper is a simple derivation from [1] and adapted to this representation learning setting. Furthermore, the proposition 2 the authors write inner product with respect to H. What is H in this case? As a matter of fact the inner product should equal to the evaluation of the CME. Could the authors elaborate in this?

- What is the computational complexity of the methods ? Given that you have more comparisons does that play a role?

- InfoNCE is hardly the best representation learning algorithm these days. Could the authors please elaborate why methods such as SimCLR?

- the results look. very promising. However I am curious to see whether the authors could also add the table for different kernels from the other 2 datasets as I can't seem to see them in the Appendix. In addition, please correct me if I am wrong, I do not see how you currently select the hyper parameters for the RBF kernel etc. How have these been selected and was the test set used? The reason I am asking this is because if the linear kernel is good enough it is questionable how much kernels even play a role.

- Are the results reported with 1 or 2 deviations?

- How do the competing methods perform as you increase/decrease the conditioning data. It would be very enlightening to see a plot where the x axis has the number of conditioning data and y axis the performance to see how your method performs much better when there is little conditioning data available. Could the authors please comment and report on this?


[1]: Ton et al 2021 https://arxiv.org/abs/1906.02236

**Summary Of The Paper:**

The paper looks into using more conditional data than standard methods by leveraging conditional mean embeddings and this weighting more datapoint when performing contrastive learning. The methodology hence allows to more comparisons (but weighted) and hence allowing the proposed method to achieve higher representational power. The idea of CME allows them to compare the conditional data (ie usually a smaller set) with the batch and then still perform contrastive learning now will the batch. They show improved performance on a variety of experiments such as fairness, representation learning and weakly supervised learning.

**Summary Of The Review:**

I believe that the paper as its merits and really like how they have seemingly incorporated CME into representation learning and obtain good performances. However, there are still important references missing which I believe reduce the novelty by quite a margin given that it s a simple derivation from [1] applied to different setting. In addition, there are some clarification on the experimental side which I would like know. **I am more than happy to increase my score if the authors are able to clarify my concerns and will be looking forward to their rebuttal.**

---

> ### Author Response · Authors · 2021-11-19
> **Response to Reviewer imU8 Part II**
>
> **(Why InfoNCE?)** Using InfoNCE we meant the InfoNCE and the InfoNCE-inspired objective functions. All learning frameworks are based on SimCLR [7], a standard high-performing method on visual contrastive learning (and shares performances very close to SOTA results at the time when we submitted this manuscript).
>
> **(Kernel Ablations)** We appreciate the suggestions, and we would kindly remind the reviewer that such ablations on kernel choices have been provided in Table 1 in Appendix. We are also grateful for the question on how the role kernel plays.
>
> First, the performance trend does change too much and is sensitive to different bandwidths for the RBF kernel. How we select the bandwidth in the RBF kernel is based on empirical results on the development set (using the standard cross-validation technique). Below we show the result of using different $\sigma^2$ in the RBF kernel. We found that using $\sigma^2 = 1$ significantly hurts performance(only 3.0%), using $\sigma^2 = 1000$ is also sub-optimal. The performances of $\sigma^2 = 10, 100, 500$ are close.
>
> | RBF $\sigma^2$ | 1 |10 | 100 | 500 | 1000 |
> |---------------|-----------------|----------------------|-----------------------------------------------|-----------------------|-----------------------|
> | CUB acccuracy  (%)          | 3.0 | 30.9           | 32.2 | 32.0                 | 24.4
>
>
> Second, we did empirically find selecting the linear kernel is good enough for our objectives in UT-Zappos and CUB datasets, although in ImageNet-100 the linear kernel is slightly off (as shown in Table 2 in the original submission and the new results below). Nevertheless, we would clarify why we want to use kernels. Our contribution is devising a kernelized approach to resolve the insufficient data problem when performing conditional contrastive representation learning (please refer to the general response in this rebuttal). The core idea of our kernelized approach is to avoid direct conditional sampling (direct conditional sampling will be challenging under the insufficient data problem) by having the conditional mean embedding as an approximation. Using the conditional mean embedding allows a kernel formulation for our objectives (Fair_CCLK/WeakSup_CCLK/HardNeg_CCLK), but it does not mean that we should definitely use non-linear kernels. Allowing a wide range of kernel choices is simply a plus for our methods.
>
> | Kernels      | CUB | ImageNet-100
> | ----------- | ----------- | -----------
> | RBF     | 32.3      | 81.8
> | Laplacian   | 32.1       | 80.2
> | Linear | 29.4  | 77.5
> | Cosine | 29.9 | 82.4
>
> **(Standard Deviation)** We report 1 deviation.
>
> **(How do the competing methods perform as you increase/decrease the conditioning)**
>
> **[Figure link](https://i.imgur.com/PgfI7mZ.png)**
>
> Thank you for the great question. We provide the figure above, where the x-axis is the averaged number of data samples per discretized conditioning variable (cluster). The dataset is UT-Zappos and the conditioning variable is the annotative attributes. The discretization is done by grouping instances that share the same annotative attributes to the same cluster. The blue line is WeaklySup_InfoNCE, which requires discretized conditioning variables. The black line represents the proposed WeaklySup_CCLK which does not require discretization. As we can see from the figure, the performance of WeaklySup_InfoNCE suffers when the number of data per cluster (conditioning variable) is small, and WeaklySup_CCLK outperforms WeaklySup_InfoNCE in all cases. From this example, we can see that when the data is very insufficient (towards the origin in this figure), the proposed WeaklySup_CCLK outperforms WeaklySup_InfoNCE significantly.
>
> [1] Noise contrastive meta-learning for conditional density estimation using kernel mean embeddings. In International Conference on Artificial Intelligence and Statistics (pp. 1099-1107). PMLR.
>
> [2] Kernel embeddings of conditional distributions: A unified kernel framework for nonparametric inference in graphical models. Signal Processing'13.
>
> [3] Noise-contrastive estimation: A new estimation principle for unnormalized statistical models. ICML'10.
>
> [4] Integrating Auxiliary Information in Self-supervised Learning. arXiv preprint arXiv:2106.02869.
>
> [5] Conditional Contrastive Learning: Removing Undesirable Information in Self-Supervised Representations. arXiv preprint arXiv:2106.02866.
>
> [6] Contrastive learning with hard negative samples. arXiv preprint arXiv:2010.04592.
>
> [7] A simple framework for contrastive learning of visual representations. In International conference on machine learning (pp. 1597-1607). PMLR.

---

> > ### Comment · Reviewer_imU8 · 2021-11-29
> > **Thanks for the reply**
> >
> > 1. (Discussion with the Related Work [1]): [1] does compute the similarities between $\langle \phi(x), \mu_{x|z} \rangle$ but the thing is that the same is done for the noise which is in your case just $y$ i.e the contrasting quantity. This means that the it is the same with the difference being that you use the batch for contrastive learning whereas [1] uses noise. Please refer to how [1] is actually using the noise.
> >
> > 2. So $\mathcal{H}$ is the RKHS with $\phi$ as feature map? So you are saying that $\phi_{g_{\theta}(x)}$ and the CME live in the same RKHS? Which means that you are supposedly evaluating the CME no? Could you elaborate what  $\phi_{g_{\theta}(x)}$  is supposed to represent? The definition of evaluating a CME is  $ \mu_{y|z}(y) =  \langle \phi_y(.), \mu_{y|z} \rangle$
> >
> > 3. (Why InfoNCE?)  My point was more like, does your method still work in the SimCLR framework? or are the improvements incremental in this setting? InfoNCE is barely used given that there are many better options. Hence my question was more directed at whether your improvements of your proposed method would carry over to SimCLR or whether it will be washed out?
> >
> > 4. (Kernel Ablations) Thanks sorry I missed this
> >
> > 5. (How do the competing methods perform as you increase/decrease the conditioning): I am confused with this figure. How come your propped method is unaffected by the number of clusters? Your method looks at the problem where we have little conditioning data and hence having to use CME to weight the datapoint accordingly. My point was when do we not have to do this anymore. i.e. when do we have enough conditioning data? a plot where we have on the x-axis the number of conditioning data would be useful. Correct me if I am wrong but is that figure supposed to show this and if yes, how come your method is unaffected by the number of conditionings (why doesn't it get better as you increase ?)
> >
> > 6. (Standard Deviation)  If you only report 1 std then the results in Table 3 has **NO STATISTICAL DIFFERENCE with FAIR INFO NCE** I agree though that your method does not require discretisation which is good.  In Table 2 for Imagenet100 **you do not have statistical significance**. In Table 4 Cifar 10, **you do not have statistical significance**. I do not understand how the authors can bold their own numbers when there is literally no reason to.
> >
> > Please clarify the above and I would be happy to increase my score as I believe this paper to be interesting but the above concerns still make me doubt the method.

---

> > > ### Author Response · Authors · 2021-11-29
> > > **Further response to Reviewer imU8 Part II**
> > >
> > > **(InfoNCE/SimCLR)** SimCLR is a framework and InfoNCE is an objective function, and SimCLR uses InfoNCE as its objective function. Our work considers SimCLR as our base framework. We suppose the confusion comes from the following: the paper [2] first proposed InfoNCE loss within the contrastive predictive coding (CPC) framework. We do not consider the CPC framework but the SimCLR framework.
> > >
> > > **(The number of conditioning.)** We believe that we misinterpret the wording "the number of conditioning". Now suppose "the number of conditioning" you mean is the "the number of clusters". We first like to mention that the conditional variable $Z$ in the experiment in our plot are annotative attributes, which do not come in the form of clusters but vector-values. Our method (WeakSup_CCLK) directly works with "vector-values" conditional variables, while prior work (WeakSup_InfoNCE) can only work with conditional variables with cluster forms. Hence, it is a need to perform clustering on top of the "vector-values" conditional variable to make it suitable for WeakSup_InfoNCE. Our plot is showing that the number of clusters for the conditional variable will affect the performance of WeakSup_InfoNCE. And because our method can directly work with "vector-values" conditional variables, the performance is unaffected. Please let us know if you are expecting other kinds of experiments, and we will try to include them before the rebuttal period ends (which is today).
> > >
> > > **(Statistical Difference)** We are sorry for the bold. We will unbold the numbers as suggested by the reviewer in the final manuscript, as of now we are not able to update the manuscript.
> > >
> > > [1] Noise contrastive meta-learning for conditional density estimation using kernel mean embeddings. In International Conference on Artificial Intelligence and Statistics (pp. 1099-1107). PMLR.
> > >
> > > [2] Representation Learning with Contrastive Predictive Coding.

---

> > > > ### Comment · Reviewer_imU8 · 2021-12-03
> > > > **Thanks for the clarifications**
> > > >
> > > > - Thanks for clarifying
> > > > - Evaluating a CME is literally the inner product between an element in the RKHS and the CME. Your $\phi(g_{\theta_x}(x))$ is an element in the RKHS and hence you are evaluating the CME. I do note that in your case what you are doing is $y*=g_{\theta_x}(x)$ then map it into the correct RKHS (same as CME). Hence this is different but pretty much the same objective as in [1]. I do agree though that the application is much different and hence has merit.
> > > > - InfoNCE/SimCLR: thanks for clarification. Sorry i missed that.
> > > > - The number of conditioning: By number of conditioning i meant the problem that you are trying to solve, which is the case when you dont have enough conditioning data and hence have to resort to a weighting.
> > > > - Statistical Difference: So that are admitting that you method does not actually perform better in most experiments then? Out of 6 experiments yours is only better in the weakly_sup on UT Zappos,CUB and ImageNet100 hardneg  if i read correctly?
> > > >
> > > > Overall, i like the idea and the additional clarification make me raise my score. However, I still have reservations in the way the results were reported. As well as the ablation study on the number of conditioning.

---

> > > > > ### Author Response · Authors · 2021-12-03
> > > > > **Follow-up**
> > > > >
> > > > > Thanks for further discussions and comments. We will note in our revised manuscript that our methods are not significantly better than baselines if the numbers do not differ more than two standard deviations.

---

> > > ### Author Response · Authors · 2021-11-29
> > > **Further response to Reviewer imU8 Part I**
> > >
> > > We are happy to provide further responses to address the comments from the reviewer, and please do let us know if there are more questions.
> > >
> > > **(Discussion with Related Work [1])** We agree with the reviewer that, objective-wise, we use different contrasting quantities: [1] computes $\langle \phi(x), \mu_{X|z} \rangle$ and ours computes $\langle \phi(x), \mu_{Y|z} \rangle$. Note that this is the statement we made in our prior response when we discuss the difference between objective functions. To be clear, we are not strongly defending that "the usage of different contrasting quantities makes our work completely novel". And thanks to the reviewer, we have added the discussion with this missing reference [1] into our updated manuscript. Last, we hope the contribution of our paper can be recognized: we aim to resolve a practical research challenge when performing conditional contrastive representation learning. Our application is very different from performing density estimation, which is the major application of the NCE-related approaches.
> > >
> > > **($\mathcal{H}$ and $\phi$)** Thanks for bringing more detailed questions, and please see our bullet-points-answers. We hope our response can resolve potential confusion of our method/notations.
> > >
> > > 1. *($\mathcal{H}$ is the RKHS with $\phi$ as feature map?)* Yes.
> > > 2. *($\phi\big(g_{\theta_X}(x)\big)$ and the CME live in the same RKHS?)* Our method computes $\bigg\langle \phi\Big(g_{\theta_X}(x)\Big),\mu_{Y|z} \bigg\rangle_{\mathcal{H}} = \bigg\langle \phi\Big(g_{\theta_X}(x)\Big), \Phi_Y^\top (K_Z + \lambda {\bf I})^{-1} \Gamma_Z  \gamma(z) \bigg\rangle_{\mathcal{H}}$. $\phi\Big(g_{\theta_X}(x)\Big)$ and $\Phi_Y^\top (K_Z + \lambda {\bf I})^{-1} \Gamma_Z  \gamma(z)$ live in the same RKHS. In particular, $\Phi_Y = \big[\phi\big(g_{\theta_Y}(y_1)\big), \cdots, \phi\big(g_{\theta_Y}(y_b)\big)\big]^\top$ and hence $\Phi_Y^\top (K_Z + \lambda {\bf I})^{-1} \Gamma_Z  \gamma(z)$ has the feature map $\phi$ as well.
> > > 3. *(Evaluating the CME or not?)* We do not know exactly what does it mean by evaluating the CME, but according to the reviewer's definition: evaluating the CME means computing $\bigg\langle \phi\Big(g_{\theta_Y}(y)\Big),\mu_{Y|z} \bigg\rangle_{\mathcal{H}} = \bigg\langle \phi\Big(g_{\theta_Y}(y)\Big), \Phi_Y^\top (K_Z + \lambda {\bf I})^{-1} \Gamma_Z  \gamma(z) \bigg\rangle_{\mathcal{H}}$, we do not compute this quantity in our work. We compute $\bigg\langle \phi\Big(g_{\theta_X}(x)\Big),\mu_{Y|z} \bigg\rangle_{\mathcal{H}}$instead. The difference is between using $\phi\Big(g_{\theta_Y}(y)\Big)$ or using $\phi\Big(g_{\theta_X}(x)\Big)$.
> > > 4. *(What are $\phi\Big(g_{\theta_X}(x)\Big)$ and $\phi\Big(g_{\theta_Y}(y)\Big)$?)* $x$ is the input data, and $g_{\theta_X}(x)$ denotes the represention after feeding $x$ into the mapping $g_{\theta_X}(\cdot)$. Our work considers deep neural networks for $g_{\theta_X}(\cdot)$. $\phi\Big(\cdot\Big)$ then projects the deep neural network representations $g_{\theta_X}(\cdot)$ into the RKHS. On the other hand, $Y/y$ can represent a different set of inputs. While in our work, we consider $X/Y$ from the same set of inputs.

---

> ### Author Response · Authors · 2021-11-19
> **Response to Reviewer imU8 Part I**
>
> We thank the reviewer for the comments. Please see the following response. Please let us know if any of the concerns are not addressed by our response.
>
> **(Discussion with the Related Work [1])** We thank the reviewer for suggesting the related work [1], and we are happy to elaborate on the differences between [1] and ours. We have also included the following responses in the updated manuscript.
>
> First, the goals of the two work are very different. [1] studies conditional density estimation, and our work aims to resolve a practical research challenge when performing conditional contrastive representation learning (please refer to general response in this rebuttal). Considering the large discrepancy between the research topics, we humbly argue that it is unfair to state that our work is a direct extension or derivation from [1]. The similarity between the two works is that we both use kernel conditional embedding operator [2] as an intermediate tool and are related to noise contrastive estimation [3] (our work relates to conditional InfoNCE method, the conditional InfoNCE method relates to InfoNCE, and the InfoNCE method relates to noise contrastive estimation [3]).
>
> Second, we would like to be clear on the differences between objective function designs. We define $X$ (and $Y$) as the input data, $Z$ as the conditioned variable, $\mu_{Y|Z=z}$ as the conditional mean embedding, $\phi(\cdot)$ as the feature mapping, and $\langle , \rangle$ as the inner product in RKHS. [1] presents to modify the original scoring function in noise contrastive estimation into $\langle  \phi(x) , \mu_{X|Z=z} \rangle$, which stands for an approximation of the scoring function with the data sampled from $P_{XZ}$ (see their eq. 10).  Our work presents to modify the original scoring functions in WeakSup_InfoNCE [4]/Fair_InfoNCE [5]/HardNeg_InfoNCE [6] into $\langle  \phi(x) , \mu_{Y|Z=z} \rangle$, which stands for an approximation of the scoring function with the data sampled from $P_{X|Z=z}P_{Y|Z=z}$. We kindly argue that this difference is not subtle.
>
> **(What is $\mathcal{H}$?)** Below eq. 6 in our submission, we have stated that $\mathcal{H}$ is a Reproducing Kernel Hilbert Space that is associated with the feature mapping $\phi(\cdot)$.
>
> **(The Inner Product Should Equal to the Evaluation of the CME)** We cannot understand this question properly, and hence please correct us if we do not answer it in the right direction. In Proposition 2, we are calculating the inner product $\bigg\langle \phi\Big(g_{\theta_X}(x_i)\Big), \Phi_Y^\top (K_Z + \lambda {\bf I})^{-1} \Gamma_Z  \gamma(z_i) \bigg\rangle_{\mathcal{H}}$, where the inner product is performed between feature mappings $\phi(\cdot)$s. On the other hand, computing the condtional mean embedding $\Phi_Y^\top (K_Z + \lambda {\bf I})^{-1} \Gamma_Z \gamma(z_i)$ requires computing the inner product on $Z$: $\big\langle \gamma(z_i), \gamma(z_j) \big\rangle_{\mathcal{G}}$ (the notations are defined in our Step II - Kernel Formulation in Section 2.3). The inner product considered in CME is performed between the mappings $\gamma (\cdot)$s. In short, the inner products are not the same in Proposition 2 and when computing CME.
>
> **(Computational Complexity)** We will add the analysis on computational complexity in the revised manuscript. Suppose having $n$ data, simply calculating the inverse $(K_Z+\lambda I)^{-1}$ costs $O(n^3)$ or $O(n^{2.376})$ using more efficient inverse algorithms such as Coppersmith–Winograd method. In our paper, we choose the naive inverse approach with $O(n^3)$ computational cost. Next, we would like to explain the reasons why this $O(n^3)$ computation will not be an issue for our method. The first reason is that we consider a mini-batch training to constrain the size of $n$. The second reason is that the inverse $(K_Z+\lambda I)^{-1}$ does not contain gradients, and we find the computational bottlenecks are gradients computation and their updates.

---

### Official Review · Reviewer_oJk7 · 2021-11-02

**Correctness:** 2
**Technical Novelty And Significance:** 3
**Empirical Novelty And Significance:** 3
**Recommendation:** 6
**Confidence:** 4

**Details Of Ethics Concerns:**

No Ethics Concerns

**Main Review:**

Strength:
1.	The idea of using the kernel function to address the conditional sampling procedure seems novel to me. Experimental results also verify its effectiveness in weakly supervised contrastive learning and hard-negatives contrastive learning tasks.
2.	The motivation of this paper is clear. The paper is well-written and easy to follow.
3.	Codes and data are provided for reproducibility.

Weakness:
1. Computational complexity of CCL-K is missing. I think it has a higher computational cost compared with baselines due to the computation of gram matrix and its inversion.
2. I wonder whether Fair_{CCLK} can actually remove sensitive information of conditioning variable in the learned representations. Because the calculation of $K_{X \perp Y \mid Z}$ involves $K_{Z}$, i.e., the similarities of the conditioning variable. Can the information of this conditioning variable be actually removed when it is explicitly incorporated in the loss function of Fair_{CCLK}? The experimental result of Fair_{CCLK} is not very convincing because it is not compared to other fair contrastive learning baselines, such as Fair_{InfoNCE}. And the reason why Fair_{InfoNCE} is not used as a baseline is not compelling. Similar to WeaklySup_{InfoNCE} on ImageNet-100 dataset, Fair_{InfoNCE} can still be applied in ColorMNIST dataset by converting the continuous conditioning variable to a discrete one through clustering. Or the easier way is to compare these two fair contrastive learning methods on datasets with discrete conditioning variables.
3. I notice that the loss functions of Fair_{CCLK} and HardNeg_{CCLK} are exactly the same. However, the goal of Fari_{CCLK} is to remove sensitive information of Z while the goal of HardNeg_{CCLK} is to learn better representations from hard negative samples. So I think it will be better if the paper could elaborate more about how CCLK can achieve both goals with the same loss function and discuss their differences.
4. The key of CCLK is to use kernel function to measure the similarities between samples, i.e., the definition of $w(z_j, z_i)$. I think it would be better if an ablation study could be conducted to examine the effect of this component. Specifically, we could just fix all $w(z_j, z_i)$ to a constant value, such as $1/b$, to see how the CCLK performs. It is currently unclear whether the performance improvement of CCLK comes from the kernel function or the availability of all the data. This kind of ablation study could help to verify the effective part of CCLK.


**Summary Of The Paper:**

This paper studies conditional contrastive learning tasks when there is no enough data for some values of the conditioning variable. This scenario can be problematic in the conditional sampling procedure. To mitigate this problem, this paper proposes a CCL-K model. It uses a Kernel Conditional Embedding Operator to sample from all the available data and assign a kernel similarity to each sampled data, which is based on the values of the conditioning variable. The CCL-K also extends conditional contrastive learning to deal with continuous conditioning variables.

**Summary Of The Review:**

The idea of using a kernel function to address the conditional sampling problem in contrastive learning is interesting and novel. The experimental results validate its effectiveness in some contrastive learning tasks. However, my major concern is that whether  Fair_{CCLK} can actually remove sensitive information of conditioning variables (See the second point of the Weakness in Main Review). More elaboration regarding this issue is needed. Also, Fair_{CCLK} should also compare to other fair contrastive learning methods to verify its effectiveness.

---

> ### Author Response · Authors · 2021-11-19
> **Response to Reviewer oJk7 Part II**
>
> **(Fair_CCLK and HardNeg_CCLK)** Thanks for the great question. Indeed they have the same formulation. The first difference lies in the definition of $Z$. In Fair_CCLK, $Z$ is any sensitive attribute we want to reduce its effect, while in HardNeg_CCLK, the variable $Z=g_{\theta_X}(x)$ is the embedding of the data $x$. The second difference lies in their goals. On the one hand, as discussed in the response **(Remarks on Reducing Sensitive Information)**, Fair_CCLK aims to reduce the effect of $Z$ in the learned representations. On the other hand, HardNeg_CCLK constructs harder negatives to make contrastive learning procedure even more challenging by assigning higher weights for the negatively-paired data if the data are close in the embedding space.  The related work [2] has provided theoretical analysis that having hard-to-distinguish negatively-paired data could improve representation power than having easy-to-distinguish negatively-paired data.
>
> From the above two differences, we see that although Fair_CCLK and HardNeg_CCLK have the same formulation, their use cases are very different. Fair_CCLK is used when we want to improve the fairness in the contrastive learning process, assuming access to the sensitive variable. HardNeg_CCLK is used when we want to improve the representation power in the contrastive learning process.
>
> **(Remarks on $K_Z$)** We are happy to provide the analysis when setting all entries in $K_Z \in \mathbb{R}^{b \times b}$ to be $\frac{1}{b}$, which means $K_Z = \frac{1}{b}{\bf 1}{\bf 1}^\top$. Assuming $\lambda$ is small, $K_{X \perp Y|Z} = K_{XY}(K_Z + \lambda I)^{-1} K_Z = K_{XY}(\frac{1}{b}{\bf 1}{\bf 1}^\top + \lambda I)^{-1} \frac{1}{b}{\bf 1}{\bf 1}^\top$ $=K_{XY}{\bf 1} ({\bf 1}^\top{\bf 1} + b\lambda )^{-1} {\bf 1}^\top \approx \frac{1}{b}K_{XY}{\bf 1}{\bf 1}^\top$. Plugging-in this result into our objectives and when the batch size $b$ is large enough, we can find 1) WeaklySup_CCLK collapses to the constant ${\rm log}\frac{1}{b}$ and 2) Fair_CCLK/HardNeg_CCLK specialize to InfoNCE objective.
>
>
> Our intent is not to only give the mathematical formalism. We also want to provide a high-level explanation. When considering all entries being a constant for $K_Z$, we are suggesting the conditioned variable has the same value for all the data (this makes $k(z_i, z_j)$ to be a constant $\forall i, j$). First, in WeaklySup_CCLK, $Z$ stands for weak labels of data, and WeaklySup_CCLK aims to learn dissimilar representations when data having different weak labels. If all the data have the same weak label, then WeaklySup_CCLK cannot perform representation learning, which echos the fact that WeaklySup_CCLK collapses to a constant. Second, in Fair_CCLK, $Z$ stands for the sensitive attributes, and Fair_CCLK aims to reduce its effect. If all the data have the same sensitive attribute, then it is no difference between performing Fair_CCLK or InfoNCE, which echos the fact that Fair_CCLK collapses to InfoNCE. Third, HardNeg_CCLK re-weights the negatively-paired data according to $Z$. If all the data have the same $Z$, then HardNeg_CCLK assigns equal weights to all the negatively-paired data, which echos the fact that HardNeg_CCLK collapses to InfoNCE.
>
>
> Last, as suggested by the reviewer, the construction of the kernel in $K_Z$ plays an important role in our objectives. We would like to restate that our contribution is to add flexibility when performing conditional contrastive learning with the help of the kernel conditional embedding operator (please refer to general response in this rebuttal). We argue that this flexibility is the cause of the performance improvements of our objectives over baseline methods.
>
>
>
> [1] Conditional Contrastive Learning: Removing Undesirable Information in Self-Supervised Representations. arXiv preprint arXiv:2106.02866.
>
> [2] Contrastive learning with hard negative samples. arXiv preprint arXiv:2010.04592.

---

> ### Author Response · Authors · 2021-11-19
> **Response to Reviewer oJk7 Part I**
>
> We thank the reviewer for the comments. Please see the following response. Please let us know if any of the concerns are not addressed by our response.
>
>
> **(Computational Complexity)** We will add the analysis on computational complexity in the revised manuscript. Suppose having $n$ data, simply calculating the inverse $(K_Z+\lambda I)^{-1}$ costs $O(n^3)$ or $O(n^{2.376})$ using more efficient inverse algorithms such as Coppersmith–Winograd method. In our paper, we choose the naive inverse approach with $O(n^3)$ computational cost. Next, we would like to explain the reasons why this $O(n^3)$ computation will not be an issue for our method. The first reason is that we consider a mini-batch training to constrain the size of $n$. The second reason is that the inverse $(K_Z+\lambda I)^{-1}$ does not contain gradients, and we find the computational bottlenecks are gradients computation and their updates.
>
>
> **(Remarks on Reducing Sensitive Information)** We are happy to add more remarks on why *Fair_CCLK* can potentially reduce sensitive information in the learned representations. Note that *Fair_CCLK* approximates *Fair_InfoNCE* [1], and in *Fair_InfoNCE* both positively-paired and negatively-paired data are sharing the same value of the conditioned variable $Z$. To be more precise, *Fair_InfoNCE* samples positively-paired data from $P_{XY|Z=z}$ and negatively-paired data from $P_{X|Z=z}P_{Y|Z=z}$. In other words, *Fair_InfoNCE* always fixes the value of $Z=z$ in the representation learning process, and hence the variations of the conditioned variable $Z$ will not be taken into account. From an information-theoretical perspective, the variations of the conditioned variable $Z$ stand for its information: no variations means zero information. To conclude, we approximate *Fair_InfoNCE* with *Fair_CCLK* as a contrastive learning approach that reduces the effect from the sensitive information $Z$ in the learned representations.
>
>
> **(Comparison with Fair_InfoNCE [1] in the Experiments)** We agree with the reviewer that adding the comparison with Fair_InfoNCE [1] will improve the paper. We have updated this result in the revised manuscript. As suggested by the reviewer, we convert the continuous color information in the ColorMNIST dataset into discrete values through clustering, and we have chosen five sets of clusters: 3, 5, 10, 15, or 20 clusters using K-means. We provide the results of Fair_InfoNCE below:
>
> | Number of Clusters      | Top-1 Accuracy | MSE
> | ----------- | ----------- | -----------
> | 3 **(Fair_InfoNCE)**     | 82.12       | 56.27
> | 5 **(Fair_InfoNCE)**  | 84.55       | 58.67
> | 10 **(Fair_InfoNCE)** | 85.90 | 64.91
> | 15 **(Fair_InfoNCE)** | 85.22 | 65.02
> | 20 **(Fair_InfoNCE)** | 84.23 | **65.11**
> |**Fair_CCLK**| **86.4** | 64.7
>
> First, we observe that the accuracy tops at the 10-cluster setup, while the MSE tops at the 15-cluster setup. We use the 10-cluster setup in the updated manuscript because of its highest accuracy. The accuracy of Fair_InfoNCE is not as high as our proposed Fair_CCLK, although the MSE is very close to or even surpasses Fair_CCLK slightly. We observe that for Fair_InfoNCE, if the number of discretized values of $Z$ increases, the MSE in general grows, but the accuracy peaks at 10 clusters and then declines. This suggests that $Fair_{\rm InfoNCE}$ can remove more sensitive information as the granularity of $Z$ increases, but the downstream task performance may decrease. Overall, the proposed Fair_CCLK performs slightly better than the $Fair_{\rm InfoNCE}$ baseline, and does not need clustering to discretize $Z$.

---

### Official Review · Reviewer_kZnA · 2021-11-06

**Correctness:** 4
**Technical Novelty And Significance:** 3
**Empirical Novelty And Significance:** 3
**Recommendation:** 8
**Confidence:** 4

**Main Review:**

Authors proposes a solution to a very important problem. I will not question neither the motivation nor the importance of the problem. Paper is well written and mostly connects previous art. However, I believe some relations to needs be build with kernel literature (I will give examples below).
Although in general I am positive about the paper, I have some concerns:
-	Authors rely on side information which they assume is a part of the training data. In many practical cases, the cause of any bias/unfairness in data is caused by human annotation. Image annotations (cited by authors) may contain sexist/racist information. How does proposed method deal with such problems?
-	Authors conditions on z_i (please see Step – 1 Problem Setup) and develops a kernel and use it as similarity function. Indeed, one can see each z_i as a domain and K_z as a probability on densities. Please see the following the following paper
[1] Domain Generalization by Marginal Transfer Learning. JMLR, 2021. Please section 5.1.
The proposed kernel in the paper is a simplified version of kernels described in Section 5.1 of [1]. It is clear that [1] is not considering neural networks. However, a discussion is needed because of the following reason:
-	Given that authors are learning ‘a fair’ representation from data, I would like to understand the reason of fairness. If authors would have explicitly assumed a model than it is clear that ‘fairness’ is coming from the model. However, in the current version, I am not sure the whether the observed performance is because of ‘fairness’. It is very well possible that implicitly modelling problem as a domain generalization problem played a significant role.
Moreover, the final version of the used kernel has similarities with
[2] Multi-Task Learning for Contextual Bandits (Neurips 2017, I will use the arxiv version https://arxiv.org/pdf/1705.08618.pdf). Please see the first eq on page 15.
Again [2] is in the space of kernel methods however taking into account multiple domains (or multiple tasks) and defining a kernel on these tasks seems improving the baselines.
Would you please discuss the connection of the proposed method with [1] and [2]? Moreover, would you please elaborate why the proposed methods is limiting the undesired information?


**Summary Of The Paper:**

Authors proposed to use nonparametric methods for sampling data point for contrastive learning. The main concern of the paper is to limit extraction of any undesired information from the data.

**Summary Of The Review:**

Overall, I like the paper. However, I have several concerns or confusions. I am looking forward to author responses and engagement such that I can keep my score or increase it.

(After the rebuttal, I have decided to increase my score)

---

> ### Author Response · Authors · 2021-11-19
> **Response to Reviewer kZnA Part II**
>
> **(Remarks on Reducing Sensitive Information)** We are happy to add more remarks on why our objective can potentially reduce sensitive information in the learned representations. In our submission, the objective that acts to reduce the sensitive information is *Fair_CCLK*, where *Fair_CCLK* approximates *Fair_InfoNCE* [4]. In *Fair_InfoNCE*, both positively-paired and negatively-paired data are sharing the same value of the conditioned variable $Z$. To be more precise, *Fair_InfoNCE* samples positively-paired data from $P_{XY|Z=z}$ and negatively-paired data from $P_{X|Z=z}P_{Y|Z=z}$. In other words, *Fair_InfoNCE* always fixes the value of $Z=z$ in the representation learning process, and hence the variations of the conditioned variable $Z$ will not be taken into account. From an information-theoretical perspective, the variations of the conditioned variable $Z$ stand for its information: no variations means zero information. To conclude, we approximate *Fair_InfoNCE* with *Fair_CCLK* as a contrastive learning approach that reduces the effect from the sensitive information $Z$ in the learned representations.
>
> **(Contribution)** We would like to restate that our contribution is to resolve a practical research challenge when performing conditional contrastive learning (please refer to the general response in this rebuttal). We particularly thank the reviewer for suggesting related work on multi-domain and multi-task representation learning.
>
> [1] Kernel embeddings of conditional distributions: A unified kernel framework for nonparametric inference in graphical models. Signal Processing'13.
>
> [2] Domain generalization by marginal transfer learning. arXiv preprint arXiv:1711.07910.
>
> [3] Multi-task learning for contextual bandits. arXiv preprint arXiv:1705.08618.
>
> [4] Conditional Contrastive Learning: Removing Undesirable Information in Self-Supervised Representations. arXiv preprint arXiv:2106.02866.

---

> > ### Comment · Reviewer_kZnA · 2021-11-22
> > **Response to Author's Rebuttal**
> >
> > I thank the authors for their comprehensive answer and their engagement.  Most of my concerns/confussions have been resolved. I will reflect these improvment to my score.

---

> > > ### Author Response · Authors · 2021-11-22
> > > **Response to Reviewer kZnA**
> > >
> > > Dear Reviewer kZnA, thank you very much. Your update is sincerely appreciated.

---

> ### Author Response · Authors · 2021-11-19
> **Response to Reviewer kZnA Part I**
>
> We thank the reviewer for the comments. Please see the following response. Please let us know if any of the concerns are not addressed by our response.
>
> **(Auxiliary Attribute (Image Annotations) can also Contain Sensitive Information)** To address the concern from the reviewer, we now consider the case when we like to deploy our method on human data, but the side information contains both sensitive information (e.g., gender label) and non-sensitive information (e.g., occupation of a person). Our goal is to reduce the sensitive information and include the non-sensitive information in the learned representations. Note that our work devised different objectives for distinct purposes: *WeaklySup_CCLK* to *include* auxiliary information into the learned representations, and *Fair_CCLK* to *exclude* sensitive information from the learned representations. Then, we can deploy the two objectives together: *WeaklySup_CCLK* when selecting non-sensitive information as its conditioned variable and *Fair_CCLK* when selecting sensitive information as its conditioned variable.
>
> **(Remarks on Kernel)** We make three points to address the concerns about kernel.
>
> First, we agree with the reviewer that the construction of the kernel $K_Z$ relates to modeling the probability density on $Z$. In fact, this is how the kernel conditional embedding operator is derived [1].
>
> Second, we would like to comment on the following statement from the reviewer: "our proposed kernel is a simplified version of [2]". Note that there are various ways to construct kernels, and in our work, we do not propose any new kernels, instead, we consider common kernels such as RBF or consine kernel. Additionally, the kernel described in Section 5.1 in [2] is defined on a joint space, whereas the kernel in our work is defined on only one sample space. Hence, we find the usage of kernels is very different between our work and [2].
>
> Third, both the first equation on page 15 in [3] and the kernel conditional embedding operator [1] (which is used in our work) consider associative property of matrix multiplication: $A^T (A A^\top + \lambda I)^{-1}=( A^\top A + \lambda I)^{-1} A^\top$. This is the reason why the equation on page 15 in [3] has a resemblance with our objective function.
>
>
> **(Discussion with Work Learning from Multiple Domains [2] and Tasks [3])** We thank the reviewer to bring up this great question and the response will be included in the revised manuscript. The idea of treating each $z_i$ as a domain or a task indicator is fantastic and has been explored in one of our related work [4]. In specific, [4] considers a conditional contrastive learning setup, and one of its tasks is performing contrastive learning from data across multiple domains (and it considers the conditioned variable $z_i$ as the domain indicator). [4] suggests that conditioning on domain indicators can reduce domain-specific information for better generalization. Hence, we believe there is a strong connection between conditional contrastive learning (our work and [4]) and domain/task generalization [2,3]. Last, we would like to point out a future extension from our work when training on multiple domains or tasks. Not only treating $z_i$ as a domain or a task indicator, but we can also further define the kernel $k(z_i, z_j)$ as the affinity between domains or tasks. This way allows us to take the similarities and dissimilarities between domains or tasks into account when learning representations.

---

### Author Response · Authors · 2021-11-19
**General response**

We thank all the reviewers for their comments and concerns. We have provided separate responses to individual reviewers to address the concerns. We have also updated our manuscript with changes highlighted in red.

**(Research Challenge and Our Contribution)** Since some of the reviewers have misplaced our contribution, we would like to provide a general response to it.

Our submission focuses on a recently-introduced topic *"conditional contrastive learning"*, which has been applied in weakly-supervised learning [1], fair representation learning [2], and unsupervised representation learning [3]. Different from ordinary contrastive learning, conditional contrastive learning performs conditional sampling (sampling the data when conditioned on a particular variable) to construct positively- or negatively-pairs. The novelty of this work is resolving a research challenge in conditional contrastive learning: insufficient data problem. For example, [1] construct positively-paired data when the data are sharing the same weak label (the weak label is the conditioned variable), such as sampling the health data from $90$-yrs old people. In this example, "age" is the conditioned variable and "$90$-yrs old" is its condioned value. Nonetheless, performing such conditional sampling assumes that our dataset contains enough health data from $90$-yrs old people. What if there is only a small number of health data from $90$-yrs old people? This scenario prevents us to perform effective conditional sampling.

Prior conditional sampling literature does not discuss the insufficient data problem, and their approaches cannot be applied under this practical challenge. Our submission tackles this problem by approximating the conditional sampling estimation via the well-established trick - *kernel conditional embedding operator* from the kernel literature. To be clear, our contribution lies in resolving the research challenge of the insufficient data problem in conditional contrastive learning.


[1] Integrating Auxiliary Information in Self-supervised Learning. arXiv preprint arXiv:2106.02869.

[2] Conditional Contrastive Learning: Removing Undesirable Information in Self-Supervised Representations. arXiv preprint
arXiv:2106.02866.

[3] Contrastive learning with hard negative samples. arXiv preprint arXiv:2010.04592.

---

### Decision · Program_Chairs · 2022-01-20

**Decision:**

Accept (Poster)

**Comment:**

The reviewers all acknowledge the importance of the paper as it addressed the challenge of the insufficient data problem in conditional contrastive learning, feeling that the idea was novel, the experiments verified the effectiveness of the model well, and the paper is well written. Reviewers also raised some good questions, such as the computational complexity, comparison with Fair_InfoNCE in the experiments, and kernel ablations. These questions are well addressed in the rebuttal and the revised version. One reviewer raised the issue of similarity to [1]. After taking a close look at this paper and [1], the AC felt that the motivation and focus of this paper are quite different from [1]. The authors should incorporate all the rebuttal info into the final version.

[1] Jean-Francois Ton et al. 2021.